# Training Unbiased Diffusion Models From Biased Dataset

**Yeongmin Kim**[1][*]**, Byeonghu Na**[1]**, Minsang Park**[1]**, JoonHo Jang**[1]**, Dongjun Kim**[1]**,
Wanmo Kang**[1]**, Il-Chul Moon**[1,2]

## Abstract

With significant advancements in diffusion models, addressing the potential risks of dataset bias becomes increasingly important. Since generated outputs directly suffer from dataset bias, mitigating latent bias becomes a key factor in improving sample quality and proportion. This paper proposes time-dependent importance reweighting to mitigate the bias for the diffusion models. We demonstrate that the time-dependent density ratio becomes more precise than previous approaches, thereby minimizing error propagation in generative learning. While directly applying it to score-matching is intractable, we discover that using the time-dependent density ratio both for reweighting and score correction can lead to a tractable form of the objective function to regenerate the unbiased data density. Furthermore, we theoretically establish a connection with traditional score-matching, and we demonstrate its convergence to an unbiased distribution. The experimental evidence supports the usefulness of the proposed method, which outperforms baselines including time-independent importance reweighting on CIFAR-10, CIFAR-100, FFHQ, and CelebA with various bias settings. Our code is available at `https://github.com/alsdudrla10/TIW-DSM`.

## 1 Introduction

Recent developments on diffusion models (Song et al., 2020; Ho et al., 2020) make it possible to generate high-fidelity images (Dhariwal & Nichol, 2021; Kim et al., 2023), and dominate generative learning frameworks. The diffusion models deliver promising sample quality in various applications, i.e. text-to-image generation (Rombach et al., 2022; Nichol et al., 2022), image-to-image translation (Meng et al., 2021; Zhou et al., 2024), and counterfactual generation (Kim et al., 2022b; Wang et al., 2023a). As diffusion models become increasingly prevalent, addressing the potential risks on its *dataset bias* becomes more crucial, which had been less studied in the generative model community.

The dataset bias is pervasive in real world datasets, which ultimately affects the behavior of machine learning systems (Tommasi et al., 2017). As shown in Figure 1a, there exists a bias in the sensitive attribute in the CelebA (Liu et al., 2015) benchmark dataset. In generative modeling, the statistics of generated samples are directly influenced or even exacerbated by dataset bias (Hall et al., 2022; Frankel & Vendrow, 2020). The underlying bias factor is often left unannotated (Torralba & Efros, 2011), so it is a challenge to mitigate the bias in an unsupervised manner. Importance reweighting is one of the standard training techniques for de-biasing in generative models. Choi et al. (2020) propose pioneering work in generative modeling by utilizing a pre-trained density ratio between biased and unbiased distributions. However, the estimation of density ratio is notably imprecise (Rhodes et al., 2020), leading to error propagation in training generative models.

We introduce a method called Time-dependent Importance reWeighting (TIW), designed for diffusion models. This method estimates the time-dependent density ratio between the perturbed biased distribution and the perturbed unbiased distribution using a time-dependent discriminator. We investigate the perturbation provides benefits for accurate estimation of the density ratio. We introduce that the time-dependent density ratio can serve as a weighting mechanism, as well as a score correction. By utilizing these dual roles by density ratios, simultaneously; we render the objective function tractable and establish a theoretical equivalence with existing score-matching objectives from unbiased distributions.

---

[*]Correspondence to Yeongmin Kim ⟨alsdudrla10@kaist.ac.kr⟩,[1]KAIST, [2]Summary.AI

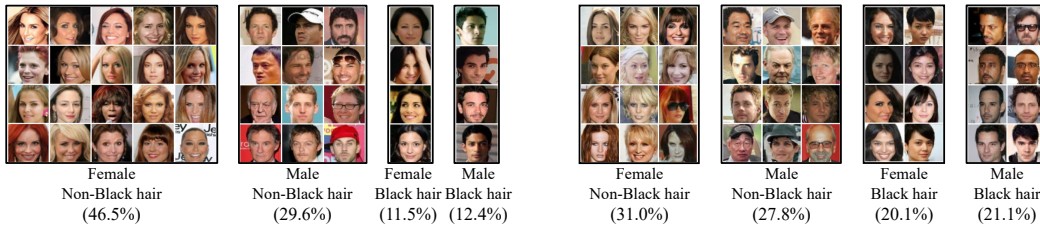

|  (a) CelebA benchmark dataset | (b) Generated samples from proposed method |

Figure 1: The samples that reflect the proportion of four latent subgroups. The proposed method mitigates the latent bias statistics as shown in (b).

We test our method on the CIFAR-10, CIFAR-100 (Krizhevsky, 2009), FFHQ (Karras et al., 2019), and CelebA datasets. We observed our method outperforms the time-independent importance reweighting and naive baselines in various bias settings.

## 2 BACKGROUND

### 2.1 PROBLEM SETUP

The goal of generative modeling is to estimate the underlying true data distribution $p_{\text{data}} : \mathcal{X} \to \mathbb{R}_{\geq 0}$, so this distribution enables likelihood evaluations and sample generations. In this process, we often consider an observed sample dataset, $\mathcal{D}_{\text{obs}} = \left\{ \mathbf{x}^{(1)}, ..., \mathbf{x}^{(n)} \right\}$ with i.i.d. sampling of $\mathbf{x}^{(i)}$, from $p_{\text{data}}$ to be unbiased with respect to the underlying latent factors, but this is not true if the sampling procedure is biased. $\mathcal{D}_{\text{obs}}$ could be biased due to social, geographical, and physical factors resulting in deviations from the intended purposes. Subsequently, the parameter $\boldsymbol{\theta}$ of the modeled distribution $p_{\boldsymbol{\theta}} : \mathcal{X} \to \mathbb{R}_{\geq 0}$ also becomes biased, which won't converge to $p_{\text{data}}$ through learning on $\boldsymbol{\theta}$ with $\mathcal{D}_{\text{obs}}$.

Building upon prior research (Choi et al., 2020), we assume that the accessible data $\mathcal{D}_{\text{obs}}$ consists of two sets: $\mathcal{D}_{\text{obs}} = \mathcal{D}_{\text{bias}} \cup \mathcal{D}_{\text{ref}}$. The elements in $\mathcal{D}_{\text{bias}}$ are i.i.d. samples from an unknown biased distribution $p_{\text{bias}} : \mathcal{X} \to \mathbb{R}_{\geq 0}$. Note that $p_{\text{bias}}$ deviates from $p_{\text{data}}$ because of its unknown sampling bias. Each element of $\mathcal{D}_{\text{ref}}$ is i.i.d. sampled from $p_{\text{data}}$, but $|\mathcal{D}_{\text{ref}}|$ is relatively smaller than $|\mathcal{D}_{\text{bias}}|$. We also follow a weak supervision setting, which does not provide explicit bias in $p_{\text{bias}}$; but we assume that the origin of data instances is known to be either $\mathcal{D}_{\text{ref}}$ or $\mathcal{D}_{\text{bias}}$.

### 2.2 DIFFUSION MODEL AND SCORE MATCHING

This paper focuses on diffusion models to parameterize model distribution $p_{\boldsymbol{\theta}}$. The diffusion model is well explained by Stochastic Differential Equations (SDEs) (Song et al., 2020; Anderson, 1982). For a data random variable $\mathbf{x}_0 \sim p_{\text{data}}$, the forward process in eq. (1) perturbs it into a noise random variable $\mathbf{x}_T$. The reverse process in eq. (2) transforms noise random variable $\mathbf{x}_T$ to $\mathbf{x}_0$.

$$d\mathbf{x}_t = \mathbf{f}(\mathbf{x}_t, t)dt + g(t)d\mathbf{w}_t, \tag{1}$$

$$d\mathbf{x}_t = [\mathbf{f}(\mathbf{x}_t, t) - g^2(t)\nabla \log p_{\text{data}}^t(\mathbf{x}_t)]d\bar{t} + g(t)d\bar{\mathbf{w}}_\mathbf{t}, \tag{2}$$

where $\mathbf{w}_t$ denotes a standard Wiener process, $\mathbf{f}(\cdot, t) : \mathbb{R}^d \to \mathbb{R}^d$ is a drift term, and $g(\cdot) : \mathbb{R} \to \mathbb{R}$ is a diffusion term, $\bar{\mathbf{w}}_\mathbf{t}$ denotes the Wiener process when time flows backward, and $p_{\text{data}}^t(\mathbf{x}_t)$ is the probability density function of $\mathbf{x}_t$. To construct the reverse process, the time-dependent score function is approximated through a neural network $\mathbf{s}_{\boldsymbol{\theta}}(\mathbf{x}_t, t) \approx \nabla \log p_{\text{data}}^t(\mathbf{x}_t)$. The score-matching objective is derived from the Fisher divergence (Song & Ermon, 2019) as described in eq. (3).

$$\mathcal{L}_{\text{SM}}(\boldsymbol{\theta}; p_{\text{data}}) := \frac{1}{2} \int_0^T \mathbb{E}_{p_{\text{data}}^t(\mathbf{x}_t)}[\lambda(t)||\mathbf{s}_{\boldsymbol{\theta}}(\mathbf{x}_t, t) - \nabla \log p_{\text{data}}^t(\mathbf{x}_t)||_2^2]dt, \tag{3}$$

$$\mathcal{L}_{\text{DSM}}(\boldsymbol{\theta}; p_{\text{data}}) := \frac{1}{2} \int_0^T \mathbb{E}_{p_{\text{data}}(\mathbf{x}_0)}\mathbb{E}_{p(\mathbf{x}_t|\mathbf{x}_0)}[\lambda(t)||\mathbf{s}_{\boldsymbol{\theta}}(\mathbf{x}_t, t) - \nabla \log p(\mathbf{x}_t|\mathbf{x}_0)||_2^2]dt, \tag{4}$$

where $\lambda(\cdot) : [0, T] \to \mathbb{R}_+$ is a temporal weighting function. However, $\mathcal{L}_{\text{SM}}$ is intractable because computing $\nabla \log p_{\text{data}}^t(\mathbf{x}_t)$ from a sample $\mathbf{x}_t$ is impossible. To make score-matching tractable, $\mathcal{L}_{\text{DSM}}$

is commonly used as an objective function. $\mathcal{L}_{\text{DSM}}$ only needs to calculate $\nabla \log p(\mathbf{x}_t | \mathbf{x}_0)$, which comes from the forward process. Note that $\mathcal{L}_{\text{DSM}}$ is equivalent to $\mathcal{L}_{\text{SM}}$ up to a constant with respect to $\boldsymbol{\theta}$ (Vincent, 2011; Song & Ermon, 2019).

## 2.3 DENSITY RATIO ESTIMATION

The density ratio estimation (DRE) through discriminative training (also known as noise contrastive estimation) (Gutmann & Hyvärinen, 2010; Sugiyama et al., 2012) is a statistical technique that provides the likelihood ratio between two probability distributions. This estimation assumes that we can access samples from two distributions $p_{\text{data}}$ and $p_{\text{bias}}$. Afterwards, we set pseudo labels $y = 1$ on samples from $p_{\text{data}}$, and $y = 0$ on samples from $p_{\text{bias}}$. The discriminator $d_{\boldsymbol{\phi}} : \mathcal{X} \rightarrow [0, 1]$, which predicts such pseudo labels, can approximate the probability of label given $\mathbf{x}_0$ through $p(y = 1 | \mathbf{x}_0) \approx d_{\boldsymbol{\phi}}(\mathbf{x}_0)$. The optimal discriminator $\boldsymbol{\phi}^* = \arg \min_{\boldsymbol{\phi}} \left[ \mathbb{E}_{p_{\text{data}}(\mathbf{x}_0)}[-\log d_{\boldsymbol{\phi}}(\mathbf{x}_0)] + \mathbb{E}_{p_{\text{bias}}(\mathbf{x}_0)}[-\log(1 - d_{\boldsymbol{\phi}}(\mathbf{x}_0))] \right]$ represents the density ratio from the following relation in eq. (5). We define $w_{\boldsymbol{\phi}^*}(\mathbf{x}_0)$ as the true density ratio.

$$w_{\boldsymbol{\phi}^*}(\mathbf{x}_0) := \frac{p_{\text{data}}(\mathbf{x}_0)}{p_{\text{bias}}(\mathbf{x}_0)} = \frac{p(\mathbf{x}_0 | y = 1)}{p(\mathbf{x}_0 | y = 0)} = \frac{p(y = 0)p(y = 1 | \mathbf{x}_0)}{p(y = 1)p(y = 0 | \mathbf{x}_0)} = \frac{d_{\boldsymbol{\phi}^*}(\mathbf{x}_0)}{1 - d_{\boldsymbol{\phi}^*}(\mathbf{x}_0)} \tag{5}$$

## 2.4 IMPORTANCE REWEIGHTING FOR UNBIASED GENERATIVE LEARNING

Choi et al. (2020) propose the importance reweighting to mitigate dataset bias. They originally conducted an experiment on GANs (Goodfellow et al., 2014; Brock et al., 2018), and there is no previous work on diffusion models with the same purpose.

Hence, the first approach would be utilizing the important reweighting for GANs in the diffusion models. In detail, the previous work pre-trains the density ratio $\frac{p_{\text{data}}(\mathbf{x}_0)}{p_{\text{bias}}(\mathbf{x}_0)} \approx w_{\boldsymbol{\phi}}(\mathbf{x}_0)$ as described in Section 2.3. The density ratio assigns a higher weight to the sample that appears to be from $p_{\text{data}}$ as described in eq. (6). The optimally estimated density ratio makes it possible to compute eq. (7). This can lead the $p_{\boldsymbol{\theta}}$ to converge to the true data distribution by utilizing the biased dataset. We call this method time-independent importance reweighting, and the derived objective in eq. (7) as importance reweighted denoising score-matching (IW-DSM).

$$\mathcal{L}_{\text{DSM}}(\boldsymbol{\theta}; p_{\text{data}}) = \frac{1}{2} \int_0^T \mathbb{E}_{p_{\text{bias}}(\mathbf{x}_0)} \left[ \frac{p_{\text{data}}(\mathbf{x}_0)}{p_{\text{bias}}(\mathbf{x}_0)} \ell_{\text{dsm}}(\boldsymbol{\theta}, \mathbf{x}_0) \right] dt \tag{6}$$

$$= \frac{1}{2} \int_0^T \mathbb{E}_{p_{\text{bias}}(\mathbf{x}_0)} \left[ w_{\boldsymbol{\phi}^*}(\mathbf{x}_0) \ell_{\text{dsm}}(\boldsymbol{\theta}, \mathbf{x}_0) \right] dt, \tag{7}$$

where $\ell_{\text{dsm}}(\boldsymbol{\theta}, \mathbf{x}_0) := \mathbb{E}_{p(\mathbf{x}_t | \mathbf{x}_0)}[\lambda(t) || \mathbf{s}_{\boldsymbol{\theta}}(\mathbf{x}_t, t) - \nabla \log p(\mathbf{x}_t | \mathbf{x}_0) ||_2^2]$.

# 3 METHOD

In this section, we present our approach for training an unbiased diffusion model with a weak supervision setting. Section 3.1 explains the motivation behind time-dependent importance reweighting. Section 3.2 explains the method in detail, which involves using a time-dependent density ratio for both weighting and score correction. Furthermore, we explore the relationship between our proposed objective and the previous score-matching objective.

## 3.1 WHY TIME-DEPENDENT IMPORTANCE REWEIGHTING?

Density ratio estimation (DRE) provides significant benefits for probabilistic machine learning (Song & Kingma, 2021; Aneja et al., 2020; Xiao & Han, 2022; Goodfellow et al., 2014). However, DRE suffers from estimation errors due to the *density-chasm* problem. Rhodes et al. (2020) state that the ratio estimation error increases when 1) the distance between two distributions is far, and 2) the number of samples from two distributions is small. The pre-trained density ratio from Section 2.4, $w_{\boldsymbol{\phi}}$, also suffers from this issue because 1) we handle real-world datasets that are in high dimensions, and 2) the number of reference data $|\mathcal{D}_{\text{ref}}|$ would be small. To address this problem, we investigate a method that involves using a time-dependent density ratio between perturbed

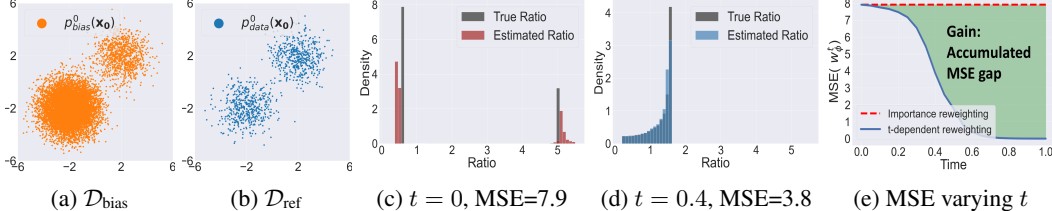

Figure 2: Accuracy of density ratio estimation between $p_{\text{bias}}$ and $p_{\text{data}}$ under diffusion process. (a-b) Samples from two distributions. (c-d) Density ratio statistics on the ground truth and the model, at each diffusion time. (e) Density ratio estimation error according to $t$. The density ratio error becomes significantly decreases as $t$ becomes larger.

distributions $p_{\text{bias}}^t(\mathbf{x}_t)$ and $p_{\text{data}}^t(\mathbf{x}_t)$. This has benefits: 1) The perturbation from the forward diffusion process makes the two distributions closer as $t$ becomes larger; and 2) the perturbation reduces Monte Carlo error in a sampling of each distribution. These two advantages of forward diffusion can contribute significantly to the accuracy of density ratio estimation.

The time-dependent density ratio $w_{\phi^*}^t(\mathbf{x}_t) := \frac{p_{\text{data}}^t(\mathbf{x}_t)}{p_{\text{bias}}^t(\mathbf{x}_t)}$ is represented by a time-dependent discriminator. We now parametrize the time-dependent discriminator $d_\phi : \mathcal{X} \times [0, T] \to [0, 1]$ which separates the samples from $p_{\text{data}}^t(\mathbf{x}_t)$ and the samples from $p_{\text{bias}}^t(\mathbf{x}_t)$. The time-dependent discriminator is optimized by minimizing temporally weighted binary cross-entropy (T-BCE) objective as described in eq. (8), where $\lambda'(t)$ denotes a temporal weighting function. We represent the time-dependent density ratio as $w_{\phi^*}^t(\mathbf{x}_t) = \frac{d_{\phi^*}(\mathbf{x}_t, t)}{1 - d_{\phi^*}(\mathbf{x}_t, t)}$.

$$\mathcal{L}_{\text{T-BCE}}(\phi; p_{\text{data}}, p_{\text{bias}}) := \int_0^T \lambda'(t) \big[ \mathbb{E}_{p_{\text{data}}^t(\mathbf{x}_t)} [-\log d_\phi(\mathbf{x}_t, t)] + \mathbb{E}_{p_{\text{bias}}^t(\mathbf{x}_t)} [-\log(1 - d_\phi(\mathbf{x}_t, t))] \big] \mathrm{d}t \tag{8}$$

Figure 2 shows the accuracy of density ratio estimation over the diffusion time interval $t \in [0, T]$, where $T = 1$. We set the 2-D distributions as follows: $p_{\text{bias}}^0(\mathbf{x}_0) := \frac{9}{10}\mathcal{N}(\mathbf{x}_0; (-2, -2)^T, \mathbf{I}) + \frac{1}{10}\mathcal{N}(\mathbf{x}_0; (2, 2)^T, \mathbf{I})$ and $p_{\text{data}}^0(\mathbf{x}_0) := \frac{1}{2}\mathcal{N}(\mathbf{x}_0; (-2, -2)^T, \mathbf{I}) + \frac{1}{2}\mathcal{N}(\mathbf{x}_0; (2, 2)^T, \mathbf{I})$. We sampled a finite number of samples from each distribution as illustrated in Figures 2a and 2b. We perturb these two distributions to $p_{\text{bias}}^t(\mathbf{x}_t)$ and $p_{\text{data}}^t(\mathbf{x}_t)$ following the Variance Preserving (VP) SDE (Ho et al., 2020; Song et al., 2020). Figures 2c and 2d illustrate the histograms of the ground truth density ratio: $w_{\phi^*}^t(\mathbf{x}_t)$, and the estimated density ratio: $w_\phi^t(\mathbf{x}_t)$, with $\mathbf{x}_t$ drawn from $\frac{1}{2}(p_{\text{bias}}^t + p_{\text{data}}^t)$. At $t = 0$, the true ratio is determined by the choice of the mode. The discriminator tends to be overconfident in favor of either $p_{\text{bias}}$ or $p_{\text{data}}$, exhibiting a skew toward either side (Figure 2c). This phenomenon is mitigated as the diffusion time increases (Figure 2d). The mean squared error (MSE) is calculated through $\mathbb{E}_{\frac{1}{2}(p_{\text{bias}}^t + p_{\text{data}}^t)}[||w_{\phi^*}^t(\mathbf{x}_t) - w_\phi^t(\mathbf{x}_t)||_2^2]$ for each time step. Figure 2e illustrates that the density ratio estimation error decreases rapidly as $t$ increases.

Applying the time-independent importance reweighting, as described in Choi et al. (2020), utilizes the density ratio only at $t = 0$ for loss computation, and this ratio becomes constant to $t$ in the score-matching. The previously discussed density-chasm creates the weight estimation error, illustrated as a red line in Figure 2e; and this error propagates through the diffusion model training. Considering the time integrating nature of score-matching objectives, the integrated estimation error of time-dependent density ratio $\int_0^1 \mathbb{E}_{\frac{1}{2}(p_{\text{bias}}^t + p_{\text{data}}^t)}[||w_{\phi^*}^t(\mathbf{x}_t) - w_\phi^t(\mathbf{x}_t)||_2^2]\mathrm{d}t$ is only 39.1%, compared to $\int_0^1 \mathbb{E}_{\frac{1}{2}(p_{\text{bias}} + p_{\text{data}})}[||w_{\phi^*}^0(\mathbf{x}_0) - w_\phi^0(\mathbf{x}_0)||_2^2]\mathrm{d}t$. We additionally discuss the benefits of time-dependent discriminator training in Appendix A.2. The natural way to reduce this DRE error is to employ time-dependent importance reweighting based on the time-dependent density ratio, as this paper suggests for the first time in the line of work on diffusion models.

### 3.2 SCORE MATCHING WITH TIME-DEPENDENT IMPORTANCE REWEIGHTING

The objective $\mathcal{L}_{\text{DSM}}$ utilizes the samples from the joint space of $p(\mathbf{x}_0, \mathbf{x}_t)$, so applying time-dependent importance reweighting is not straightforward. We start with $\mathcal{L}_{\text{SM}}$, which entails expectations on marginal distribution. We apply time-dependent importance reweighting through eq. (9).

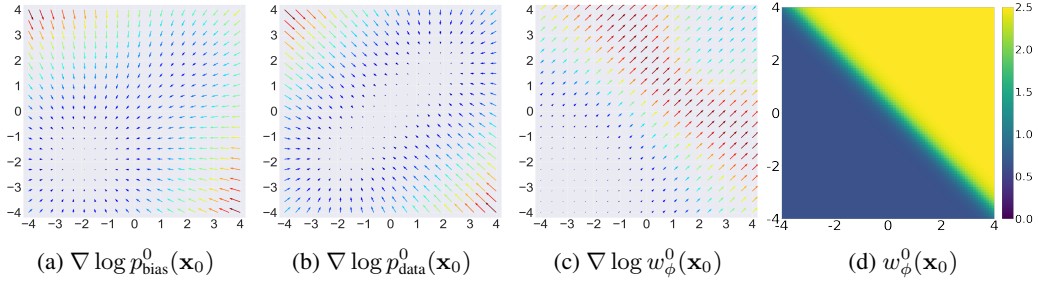

(a) $\nabla \log p_{\text{bias}}^0(\mathbf{x}_0)$     (b) $\nabla \log p_{\text{data}}^0(\mathbf{x}_0)$     (c) $\nabla \log w_\phi^0(\mathbf{x}_0)$     (d) $w_\phi^0(\mathbf{x}_0)$

Figure 3: (a-b) The score plots on $p_{\text{bias}}^0$ and $p_{\text{data}}^0$ defined in Figure 2. (c) The score plot on score correction term. (d) The reweighting value. The time-dependent density ratio simultaneously mitigates the bias through (c) and (d).

$$\mathcal{L}_{\text{SM}}(\boldsymbol{\theta}; p_{\text{data}}) = \frac{1}{2} \int_0^T \mathbb{E}_{p_{\text{bias}}^t(\mathbf{x}_t)} \left[ w_{\phi^*}^t(\mathbf{x}_t) \ell_{\text{sm}}(\boldsymbol{\theta}, \mathbf{x}_t) \right] \mathrm{d}t, \tag{9}$$

where $\ell_{\text{sm}}(\boldsymbol{\theta}, \mathbf{x}_t) := \lambda(t) \| \mathbf{s}_{\boldsymbol{\theta}}(\mathbf{x}_t, t) - \nabla \log p_{\text{data}}^t(\mathbf{x}_t) \|_2^2$, and $w_{\phi^*}^t(\mathbf{x}_t) = \frac{p_{\text{data}}^t(\mathbf{x}_t)}{p_{\text{bias}}^t(\mathbf{x}_t)}$.

Meanwhile, this objective is still intractable because we cannot evaluate $\nabla \log p_{\text{data}}^t(\mathbf{x}_t)$ from a sample $\mathbf{x}_t$. Also, there is mismatching between the sampling distribution $p_{\text{bias}}^t(\mathbf{x}_t)$ and the density function of target score $\nabla \log p_{\text{data}}^t(\mathbf{x}_t)$. This difference interferes with the straightforward conversion to a denoising score-matching approach.

To tackle this issue, we propose an objective function named time-dependent importance reweighted denoising score-matching (TIW-DSM). There exists a new score correction term, $\nabla \log w_{\phi^*}^t(\mathbf{x}_t) := \nabla \log \frac{p_{\text{data}}^t(\mathbf{x}_t)}{p_{\text{bias}}^t(\mathbf{x}_t)}$, as a regularization in the L2 loss on score-matching.

$$\mathcal{L}_{\text{TIW-DSM}}(\boldsymbol{\theta}; p_{\text{bias}}, w_{\phi^*}^t(\cdot)) \tag{10}$$
$$:= \frac{1}{2} \int_0^T \mathbb{E}_{p_{\text{bias}}(\mathbf{x}_0)} \mathbb{E}_{p(\mathbf{x}_t|\mathbf{x}_0)} \left[ \lambda(t) w_{\phi^*}^t(\mathbf{x}_t) \left[ \| s_{\boldsymbol{\theta}}(\mathbf{x}_t, t) - \nabla \log p(\mathbf{x}_t|\mathbf{x}_0) - \nabla \log w_{\phi^*}^t(\mathbf{x}_t) \|_2^2 \right] \right] \mathrm{d}t$$

Here, we briefly explore the meaning of the newly suggested regularization term through eq. (11).

$$\nabla \log w_{\phi^*}^t(\mathbf{x}_t) = \nabla \log p_{\text{data}}^t(\mathbf{x}_t) - \nabla \log p_{\text{bias}}^t(\mathbf{x}_t) \tag{11}$$

$\nabla \log w_{\phi^*}^t(\mathbf{x}_t)$ forces the model scores to move away from $\nabla \log p_{\text{bias}}^t(\mathbf{x}_t)$ and head towards $\nabla \log p_{\text{data}}^t(\mathbf{x}_t)$. Figure 3 interprets this score correction scheme on the 2-D distributions as described in Figures 2a and 2b. Figure 3a shows that $\nabla \log p_{\text{bias}}^t(\mathbf{x}_t)$ incorporates a substantial portion of the mode in the lower left. The correction term in Figure 3c exerts a force away from the biased mode, allowing the model to target the $\nabla \log p_{\text{data}}^t(\mathbf{x}_t)$ as shown in Figure 3b. Figure 3d illustrates the reweighting values, which assigns small values to the points from the biased mode, and imposes larger weights on the points from another mode. The time-dependent density ratio simultaneously mitigates the bias through score correction and reweighting.

Moving beyond the conceptual explanations, the following theorem guarantees the mathematical validity of the proposed objective function.

**Theorem 1.** $\mathcal{L}_{TIW\text{-}DSM}(\boldsymbol{\theta}; p_{bias}, w_{\phi^*}^t(\cdot)) = \mathcal{L}_{SM}(\boldsymbol{\theta}; p_{data}) + C$, where $C$ is a constant w.r.t. $\boldsymbol{\theta}$.

See Appendix A.1 for the proof. We declare that the proposed objective function is equivalent to the classical score-matching objective with $p_{\text{data}}$. Despite the equivalence, implementing only $\mathcal{L}_{\text{DSM}}$ with $\mathcal{D}_{\text{ref}}$ for our problem is not a viable option due to the limited amount of $\mathcal{D}_{\text{ref}}$ from $p_{\text{data}}$. $\mathcal{L}_{\text{DSM}}$ will suffer from Monte Carlo approximation error from limited data (See Appendix C for more details). In contrast, our objective allows for the use of biased data $\mathcal{D}_{\text{bias}}$, which has many more data points. Furthermore, the following corollary guarantees the optimality of the proposed objective.

**Corollary 2.** Let $\boldsymbol{\theta}_{TIW\text{-}DSM}^* = \arg\min_{\boldsymbol{\theta}} \mathcal{L}_{TIW\text{-}DSM}(\boldsymbol{\theta}; p_{bias}, w_{\phi^*}^t(\cdot))$ be the optimal parameter. Then $s_{\boldsymbol{\theta}_{TIW\text{-}DSM}^*}(\mathbf{x}_t, t) = \nabla \log p_{data}^t(\mathbf{x}_t)$ for all $\mathbf{x}_t, t$.

While we utilize biased datasets, the equivalence of the objective functions ensures the proper optimality. We also incorporate utilizing of $\mathcal{D}_{\text{ref}}$ for practical implementation (See Appendix A.4). In summary, we can converge our model distribution to the underlying true unbiased data distribution by utilizing all observed data.

## 4 EXPERIMENTS

This section empirically validates that the proposed method effectively operates on real-world biased datasets. We outline the experiment setups below.

**Datasets** We consider CIFAR-10, CIFAR-100, FFHQ, and CelebA datasets, which are commonly used for generative learning. Note that we access the latent bias factor only for the data construction and evaluations. To construct $\mathcal{D}_{\text{bias}}$, we consider class as a latent bias factor in CIFAR-10 and CIFAR-100. For human face datasets, we consider gender as a latent bias factor for FFHQ, and both gender and hair color for CelebA. To construct $\mathcal{D}_{\text{ref}}$, we randomly sample a subset from the entire unbiased dataset. We experiment with various numbers of $|\mathcal{D}_{\text{ref}}|$ on each dataset. See Appendix D.1 for more detailed explanations of the dataset.

**Metric** Our goal is to make the model distribution converge to an unbiased distribution. To measure this, we use the Fréchet Inception Distance (FID) (Heusel et al., 2017), which measures the distance between the distributions. We calculate the FID between 1) 50k samples from the model distribution and 2) all the samples from the entire unbiased dataset.

**Baselines** We establish three baselines for our main comparison. DSM(ref) and DSM(obs) denote the naive training of diffusion model with $\mathcal{D}_{\text{ref}}$ and $\mathcal{D}_{\text{obs}}$, respectively. IW-DSM denotes a method using time-independent importance reweighting in eq. (7), and TIW-DSM denotes our method in eq. (10). Note that both IW-DSM and TIW-DSM also incorporate the use of $\mathcal{D}_{\text{ref}}$ for our experiment (See Appendix A.4 for more details), and we always use the same experimental setting across the baselines by only varying objective functions (See Appendix D.2 for the detailed training configurations).

### 4.1 LATENT BIAS ON THE CLASS

Table 1: Experimental results on CIFAR-10 and CIFAR-100 datasets with various reference size. The reference size indicates $\frac{|\mathcal{D}_{\text{ref}}|}{|\mathcal{D}_{\text{bias}}|}$. All the reported values are the FID ($\downarrow$) between the generated samples from each method and all the samples from the entire unbiased dataset.

| Data | Bias set | CIFAR-10 (LT) | | | | CIFAR-100 (LT) | | | |
|---|---|---|---|---|---|---|---|---|---|
| | Reference size | 5% | 10% | 25% | 50% | 5% | 10% | 25% | 50% |
| Method | DSM(ref) | 16.47 | 11.56 | 10.77 | 5.19 | 21.27 | 17.17 | 15.84 | 8.57 |
| | DSM(obs) | 12.99 | 10.75 | 8.45 | 7.35 | 15.20 | 11.06 | 8.36 | 6.17 |
| | IW-DSM | 15.79 | 11.45 | 8.19 | 4.28 | 20.44 | 15.87 | 12.81 | 8.40 |
| | TIW-DSM | **11.51** | **8.08** | **5.59** | **4.06** | **14.46** | **10.02** | **7.98** | **5.89** |

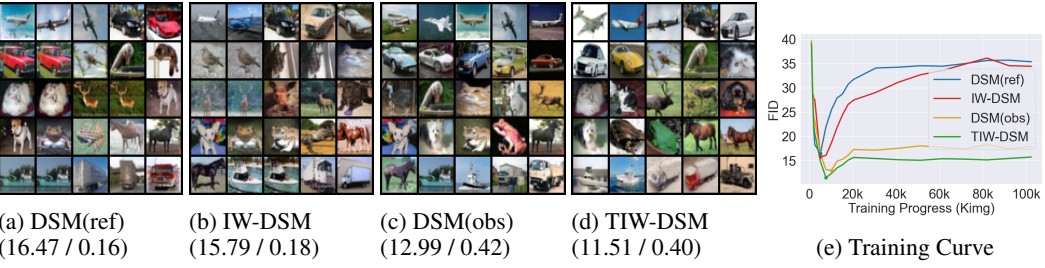

(a) DSM(ref)          (b) IW-DSM          (c) DSM(obs)          (d) TIW-DSM
(16.47 / 0.16)        (15.79 / 0.18)      (12.99 / 0.42)        (11.51 / 0.40)          (e) Training Curve

Figure 4: Analysis on CIFAR-10 (LT / 5%) experiments. (a-d) Samples that reflect the diversity and latent statistics with (FID / Recall). (e) Training curves for each method.

We construct $\mathcal{D}_{bias}$ following the Long Tail (LT) dataset (Cao et al., 2019) for CIFAR-10 and CIFAR-100. Table 1 shows the results with various reference sizes. First of all, the performance gets better as the reference size gets larger for all methods. Secondly, when comparing DSM(ref) and DSM(obs), we find that the naive use of $\mathcal{D}_{bias}$ yields better results when the reference size is too small, or the strength of bias is weak (case of CIFAR-100). However, DSM(obs) exhibits poor performance when the reference size becomes larger in the CIFAR-10 dataset. Since DSM(obs) does not guarantee to converge on the unbiased distribution, the performance is also not guaranteed under such an extreme bias setting. Third, IW-DSM consistently exhibits slightly better performance compared to DSM(ref). IW-DSM utilizes $\mathcal{D}_{ref}$ as well as $\mathcal{D}_{bias}$ with the weighted value. However, we observed that the reweighting value for $\mathcal{D}_{bias}$ is too small (will be discussed in section 4.4), which makes the effect of the $\mathcal{D}_{bias}$ marginal. In many cases, the performance of IW-DSM is even worse than the naive use of $\mathcal{D}_{bias}$. Finally, the proposed method TIW-DSM outperforms all the baseline models in every case we tested by a large margin. The comparison of IW-DSM and TIW-DSM directly indicates the effect of time-dependent importance reweighting. IW-DSM and TIW-DSM optimize two equivalent objective functions up to a constant under optimal density ratio functions (See Appendix A.3 for explanation), so the performance gain is purely from the accurate estimation of the time-dependent density ratio.

Figure 4 shows the samples in (a)-(d) and the convergence curve on each method in (e). DSM(ref) and IW-DSM illustrate extremely low sample diversity, which results in many samples being identical. DSM(obs) displayed a variety of samples, but it is heavily biased. Out of 10 latent classes, 2 latent classes accounted for 40% of the total proportion that is being calculated by a pre-trained classifier. TIW-DSM shows the diverse samples with unbiased proportions. We provide a quantitative measure of bias intensity in Figure 7. Additionally, Figure 4e shows that DSM(ref) and IW-DSM suffer from overfitting, which often occurs when training with limited data (See Appendix C for explanation). This could be evidence that IW-DSM cannot fully utilize the information from $\mathcal{D}_{bias}$.

## 4.2 LATENT BIAS ON SENSITIVE ATTRIBUTES

Table 2: Experimental results on FFHQ with various bias settings & reference size. The reference size indicates $\frac{|\mathcal{D}_{ref}|}{|\mathcal{D}_{bias}|}$. All the reported values are the FID ($\downarrow$).

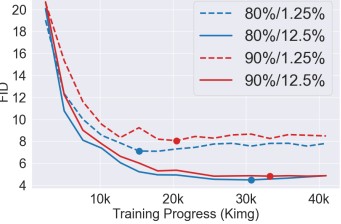

| Data | Bias set | FFHQ (80%) | | FFHQ (90%) | |
|---|---|---|---|---|---|
| | Reference size | 1.25% | 12.5% | 1.25% | 12.5% |
| Method | DSM(ref) | 12.69 | 6.22 | 12.69 | 6.22 |
| | DSM(obs) | 7.29 | 4.88 | 8.59 | 5.75 |
| | IW-DSM | 11.30 | 5.50 | 11.68 | 5.60 |
| | TIW-DSM | **7.10** | **4.49** | **8.06** | **4.83** |

Figure 5: The convergence of TIW-DSM on various bias level & reference size.

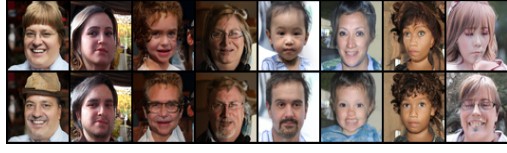

(a) FFHQ (Gender 90% / 1.25%)

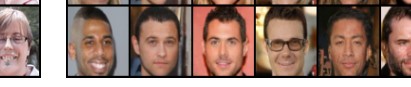

(b) CelebA (Benchmark / 5%)

Figure 6: Majority to minority conversion through our objective. The first row illustrates the samples from DSM(obs), and the second row illustrates the samples from TIW-DSM under the same random seeds. (a) indicates the female to male conversion. (b) indicates the (female & non-black hair) to (male& black hair) conversion.

We construct $\mathcal{D}_{bias}$ by making the portion of females as 80% and 90% in FFHQ experiments. Table 2 demonstrates the performance on each bias setting and various reference sizes. TIW-DSM shows superior results similar to the results from Table 1. This experiment includes a scenario with an extremely small reference set size, which is 1.25%. TIW-DSM still works well on very limited reference sizes. While TIW-DSM aims to estimate the unbiased data distribution regardless

of the intensity of bias in $\mathcal{D}_{\text{bias}}$, a lower bias intensity led to better adherence to the unbiased data distribution. Figure 5 provides the stable training curves for various experiment settings.

We also tackle the bias that actually exists in the common benchmark. We observe CelebA benchmark has suffered from bias with respect to gender and hair color. If we consider four subgroups: female without black hair ($z_{\text{F,NB}}$), male without black hair ($z_{\text{M,NB}}$), female with black hair ($z_{\text{F,B}}$), and male with black hair ($z_{\text{M,B}}$), each group has the following proportion: $p(z_{\text{F,NB}}) = 46.5\%, p(z_{\text{M,NB}}) = 29.6\%, p(z_{\text{F,B}}) = 11.5\%, p(z_{\text{M,B}}) = 12.4\%$. We construct

Table 3: Mitigating the bias exists in the CelebA benchmark dataset with 5% reference size.

| Method | FID | Latent Statistics (%) | | | |
|---|---|---|---|---|---|
| | | $z_{\text{F,NB}}$ | $z_{\text{M,NB}}$ | $z_{\text{F,B}}$ | $z_{\text{M,B}}$ |
| DSM(ref) | 2.82 | 28.0 | 29.8 | 19.3 | 22.9 |
| DSM(obs) | 3.55 | 42.8 | 30.0 | 13.0 | 14.2 |
| IW-DSM | 2.43 | 34.6 | 29.7 | 17.1 | 18.6 |
| TIW-DSM | **2.40** | 31.0 | 27.8 | 20.1 | 21.1 |

the $|\mathcal{D}_{\text{ref}}|$ as a 5% of CelebA datasets, which random samples from the unbiased dataset. Table 3 shows the experiment results for the CelebA dataset. To examine the effectiveness of weak supervision itself, we train DSM(obs) without using the information of $\mathcal{D}_{\text{ref}}$ in this experiment, which shows poor results from bias. TIW-DSM also outperforms the other baselines in terms of FID, implying that it is the best approach to address real-world bias under weak supervision. Additionally, we examine the latent statistics on generated samples using a pre-trained classifier (See Appendix D.3 for details). Figure 1b shows the generated samples from the proposed method that reflects such proportions. Figure 6 explicitly shows the reason why the bias was mitigated. Some of the samples looked in the majority latent group from DSM(obs) transformed into a minority group from TIW-DSM. These changes helped to adjust toward the equal portion in each subgroup.

## 4.3 ABLATION STUDIES

**Loss component** The proposed loss function utilizes the time-dependent density ratio for two purposes, which is the reweighting (**W**) and the score correction (**C**). We conduct ablation studies in Table 4 to assess the effectiveness of each role. Note that if we do not use both, the objective becomes the same as DSM(obs). Using only reweighting without score correction does not guarantee that the model distribution will converge to an unbiased data distribution, so the performance does not improve. While using only score correction establishes a missing link to the traditional score-matching objective, it ensures that the model converges to an unbiased data dis-

Table 4: Component ablation on the proposed method. **W** indicates the time-dependent reweighting term, **C** indicates the score correction term. All reported values are FID ($\downarrow$).

| Component | | Reference size | | | |
|---|---|---|---|---|---|
| **W** | **C** | 5% | 10% | 25% | 50% |
| ✗ | ✗ | 12.99 | 10.57 | 8.45 | 7.35 |
| ✓ | ✗ | 13.27 | 10.80 | 8.26 | 7.28 |
| ✗ | ✓ | 11.62 | 8.15 | **5.43** | 4.14 |
| ✓ | ✓ | **11.51** | **8.08** | 5.59 | **4.06** |

tribution (See Appendix A.5 for mathematical explanation), which showed quite good results. The use of these two components simultaneously performs best in most cases.

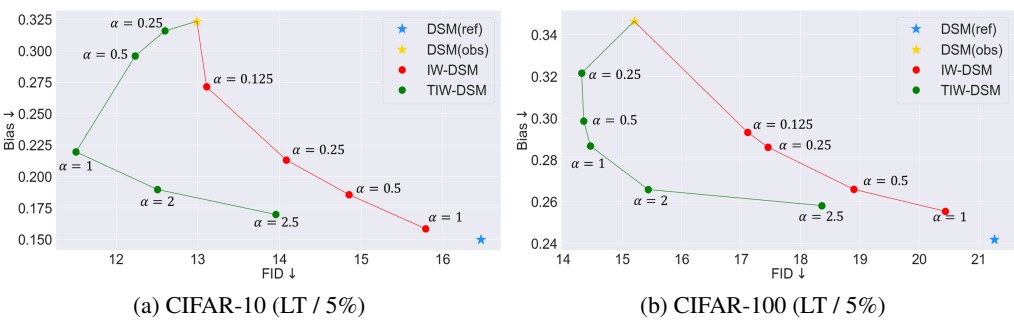

(a) CIFAR-10 (LT / 5%)      (b) CIFAR-100 (LT / 5%)

Figure 7: Bias - FID tradeoffs on the methods. We sweep $\alpha \in \{0.25, 0.5, 1, 2, 2.5\}$ for TIW-DSM, and $\alpha \in \{0.125, 0.5, 0.25, 1\}$ for IW-DSM.

**Density ratio scaling** The density ratio or confidence of the classifier can be scaled through a hyperparameter after training (Dhariwal & Nichol, 2021). We generalize our objective utilizing the $\alpha$-scaled density ratio: $\mathcal{L}_{\text{TIW-DSM}}(\boldsymbol{\theta}; p_{\text{bias}}, w^t_{\phi^*}(\cdot)^\alpha)$. Note that $\alpha = 1$ indicates the original objective function and $\alpha = 0$ becomes equivalent to DSM(obs), which is explained in Appendix A.6. We consider the experiments on CIFAR-10 and CIFAR-100 with a 5% reference set size. We also conduct $\alpha$ scaling on the IW-DSM baseline. For quantitative analyses, we also measure the strength of bias through $Bias := \Sigma_z ||\mathbb{E}_{\mathbf{x} \sim \mathcal{D}_{\text{ref}}}[p(z|\mathbf{x})] - \mathbb{E}_{\mathbf{x} \sim p_{\boldsymbol{\theta}}}[p(z|\mathbf{x})]||_2$ (See Appendix D.3 for more detail about this metric). Figure 7 illustrates that DSM(ref) shows a poor FID because it only trains on a small amount of $\mathcal{D}_{\text{ref}}$ while being free from the bias. DSM(obs) achieves better FID from a larger amount of data but suffers from bias. IW-DSM almost linearly trade-off these two metrics by adjusting $\alpha$. TIW-DSM showed improvements in both metrics within the alpha range of 0 to 1. Furthermore, TIW-DSM outperforms IW-DSM significantly in terms of FID at the same bias strength.

## 4.4 DENSITY RATIO ANALYSIS

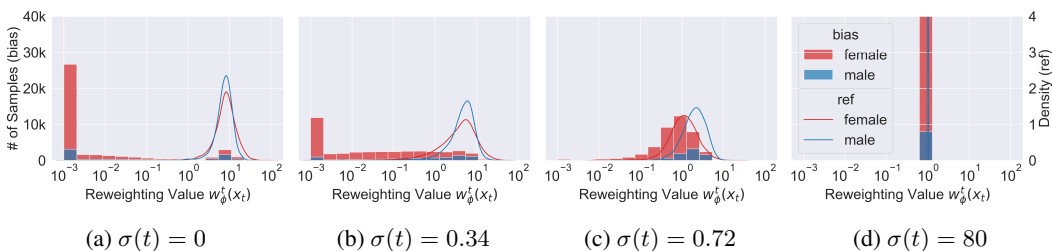

(a) $\sigma(t) = 0$      (b) $\sigma(t) = 0.34$      (c) $\sigma(t) = 0.72$      (d) $\sigma(t) = 80$

Figure 8: Reweighting value analysis on $\mathcal{D}_{\text{bias}}$ and $\mathcal{D}_{\text{ref}}$ of FFHQ (Gender 80% / 12.5%) according to diffusion time $\sigma(t)$. (a) Most of the reweighting value on $\mathcal{D}_{\text{bias}}$ is extremely small. (d) Most of the reweighting value is 1 on both $\mathcal{D}_{\text{bias}}$, and $\mathcal{D}_{\text{ref}}$. (b-c) smooth interpolation between (a) and (c).

This section investigates the importance reweighting value according to the diffusion time in our experiment. Figure 8 illustrates the histrogram of reweighting values on $\mathcal{D}_{\text{bias}}$ in FFHQ (Gender 80% / 12.5%). When the diffusion time $\sigma(t) = 0$, the trained discriminator predicts overconfidently, resulting in more than 75% of $\mathcal{D}_{\text{bias}}$ being assigned weights less than 0.01. Since IW-DSM only uses the weight value on $\sigma(t) = 0$, it does not utilize most of the information from $\mathcal{D}_{\text{bias}}$. This is the reason why the performance of IW-DSM is only marginally better than DSM(ref). While the perturbation undergoes, the reweighting value grows rapidly, which TIW-DSM leads to utilizing more information from $\mathcal{D}_{\text{bias}}$. Note that the minority latent group (or, the males in this setting) tends to get a higher value of reweighting value than the major group (female group) in each diffusion time step, which is the reason for bias mitigation. Figure 9 shows point-wise examples that change the importance weights in $\mathcal{D}_{\text{bias}}$. $\mathbf{x}^{(2)}$ and $\mathbf{x}^{(3)}$ have

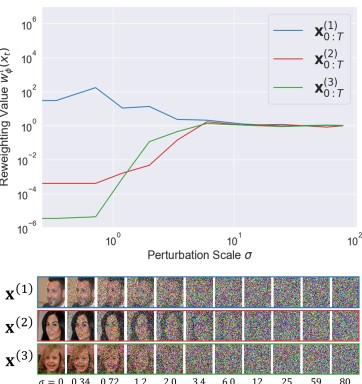

Figure 9: The density ratio changes according to the diffusion time.

extremely low reweighting values at $\sigma(t) = 0$, but these weights increase as time progresses, providing valuable information for TIW-DSM training.

## 5 CONCULSION

In this paper, we address the problem of dataset bias for the diffusion models. We highlight the previous time-independent importance reweighting undergoes error propagation from density ratio estimation, and the proposed time-dependent importance reweighting alleviates such problems. We derive the proposed weighting objective to become tractable by utilizing the time-dependent density ratio for reweighting as well as score correction. The proposed objective is connected to the traditional score-matching objective with unbiased distribution, which guarantees convergence to an unbiased distribution. Our experimental results on various kinds of datasets, weak supervision settings, and bias settings validate the proposed method's notable benefits.

ACKNOWLEDGMENTS

This research was supported by AI Technology Development for Commonsense Extraction, Reasoning, and Inference from Heterogeneous Data(IITP) funded by the Ministry of Science and ICT(2022-0-00077).

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

CONTENTS

# A    PROOFS AND MATHEMATICAL EXPLANATIONS

## A.1    PROOF OF THEOREM 1

**Theorem 1.** $\mathcal{L}_{TIW\text{-}DSM}(\boldsymbol{\theta}; p_{bias}, w_{\boldsymbol{\phi}^*}^t(\cdot)) = \mathcal{L}_{SM}(\boldsymbol{\theta}; p_{data}) + C$, where $C$ is a constant w.r.t. $\boldsymbol{\theta}$.

*Proof.* First, the score-matching objective $\mathcal{L}_{\text{SM}}(\boldsymbol{\theta}; p_{\text{data}})$ can be derived as follows.

$$\mathcal{L}_{\text{SM}}(\boldsymbol{\theta}; p_{\text{data}}) = \frac{1}{2} \int_0^T \mathbb{E}_{p_{\text{bias}}^t(\mathbf{x}_t)} \left[ w_{\boldsymbol{\phi}^*}^t(\mathbf{x}_t) \ell_{\text{sm}}(\boldsymbol{\theta}; \mathbf{x}_t) \right] dt \tag{12}$$

$$= \frac{1}{2} \int_0^T \mathbb{E}_{p_{\text{bias}}^t(\mathbf{x}_t)} \left[ w_{\boldsymbol{\phi}^*}^t(\mathbf{x}_t) \lambda(t) || \mathbf{s}_{\boldsymbol{\theta}}(\mathbf{x}_t, t) - \nabla \log p_{\text{data}}(\mathbf{x}_t) ||_2^2 \right] dt \tag{13}$$

$$= \frac{1}{2} \int_0^T \mathbb{E}_{p_{\text{bias}}^t(\mathbf{x}_t)} \left[ w_{\boldsymbol{\phi}^*}^t(\mathbf{x}_t) \lambda(t) \left[ || \mathbf{s}_{\boldsymbol{\theta}}(\mathbf{x}_t, t) ||_2^2 - 2 \nabla \log p_{\text{data}}(\mathbf{x}_t)^T \mathbf{s}_{\boldsymbol{\theta}}(\mathbf{x}_t, t) \right. \right.$$
$$\left. \left. + || \nabla \log p_{\text{data}}(\mathbf{x}_t) ||_2^2 \right] \right] dt \tag{14}$$

We further derive the inner product term in the above equation using eq. (11).

$$\mathbb{E}_{p_{\text{bias}}^t(\mathbf{x}_t)} \left[ \nabla \log p_{\text{data}}(\mathbf{x}_t)^T \mathbf{s}_{\boldsymbol{\theta}}(\mathbf{x}_t, t) \right] \tag{15}$$

$$= \int p_{\text{bias}}^t(\mathbf{x}_t) \left[ \left[ \nabla \log p_{\text{bias}}(\mathbf{x}_t) + \nabla \log w_{\boldsymbol{\phi}^*}(\mathbf{x}_t) \right]^T \mathbf{s}_{\boldsymbol{\theta}}(\mathbf{x}_t, t) \right] d\mathbf{x}_t \tag{16}$$

$$= \int p_{\text{bias}}^t(\mathbf{x}_t) \left[ \nabla \log p_{\text{bias}}(\mathbf{x}_t)^T \mathbf{s}_{\boldsymbol{\theta}}(\mathbf{x}_t, t) \right] d\mathbf{x}_t + \int p_{\text{bias}}^t(\mathbf{x}_t) \left[ \nabla \log w_{\boldsymbol{\phi}^*}(\mathbf{x}_t)^T \mathbf{s}_{\boldsymbol{\theta}}(\mathbf{x}_t, t) \right] d\mathbf{x}_t \tag{17}$$

We obtain the derivation for the first term of eq. (17) using the log derivative trick.

$$\int p_{\text{bias}}^t(\mathbf{x}_t) \nabla \log p_{\text{bias}}(\mathbf{x}_t)^T \mathbf{s}_{\boldsymbol{\theta}}(\mathbf{x}_t, t) d\mathbf{x}_t \tag{18}$$

$$= \int \nabla p_{\text{bias}}^t(\mathbf{x}_t)^T \mathbf{s}_{\boldsymbol{\theta}}(\mathbf{x}_t, t) d\mathbf{x}_t \tag{19}$$

$$= \int \left[ \nabla \int p_{\text{bias}}(\mathbf{x}_0) p_{0t}(\mathbf{x}_t | \mathbf{x}_0) d\mathbf{x}_0 \right]^T \mathbf{s}_{\boldsymbol{\theta}}(\mathbf{x}_t, t) d\mathbf{x}_t \tag{20}$$

$$= \int \left[ \int p_{\text{bias}}(\mathbf{x}_0) \nabla p_{0t}(\mathbf{x}_t | \mathbf{x}_0) d\mathbf{x}_0 \right]^T \mathbf{s}_{\boldsymbol{\theta}}(\mathbf{x}_t, t) d\mathbf{x}_t \tag{21}$$

$$= \int \int p_{\text{bias}}(\mathbf{x}_0) \nabla p_{0t}(\mathbf{x}_t | \mathbf{x}_0)^T \mathbf{s}_{\boldsymbol{\theta}}(\mathbf{x}_t, t) d\mathbf{x}_t d\mathbf{x}_0 \tag{22}$$

$$= \int \int p_{\text{bias}}(\mathbf{x}_0) p_{0t}(\mathbf{x}_t | \mathbf{x}_0) \nabla \log p_{0t}(\mathbf{x}_t | \mathbf{x}_0)^T \mathbf{s}_{\boldsymbol{\theta}}(\mathbf{x}_t, t) d\mathbf{x}_t d\mathbf{x}_0 \tag{23}$$

$$= \mathbb{E}_{p_{\text{bias}}(\mathbf{x}_0)} \mathbb{E}_{p_{0t}(\mathbf{x}_t | \mathbf{x}_0)} \left[ \nabla \log p_{0t}(\mathbf{x}_t | \mathbf{x}_0)^T \mathbf{s}_{\boldsymbol{\theta}}(\mathbf{x}_t, t) \right] \tag{24}$$

Applying eqs. (17) and (24) to eq. (14), we have:

$$\mathcal{L}_{\text{SM}}(\boldsymbol{\theta}; p_{\text{data}}) \tag{25}$$

$$= \frac{1}{2} \int_0^T \mathbb{E}_{p_{\text{bias}}^t(\mathbf{x}_t)} \left[ w_{\boldsymbol{\phi}^*}^t(\mathbf{x}_t) \lambda(t) \left[ ||\mathbf{s}_{\boldsymbol{\theta}}(\mathbf{x}_t, t)||_2^2 - 2\nabla \log p_{\text{data}}(\mathbf{x}_t)^T \mathbf{s}_{\boldsymbol{\theta}}(\mathbf{x}_t, t) \right. \right.$$
$$\left. \left. + ||\nabla \log p_{\text{data}}(\mathbf{x}_t)||_2^2 \right] \right] \mathrm{d}t \tag{26}$$

$$= \frac{1}{2} \int_0^T \mathbb{E}_{p_{\text{bias}}(\mathbf{x}_0)} \mathbb{E}_{p_{0t}(\mathbf{x}_t|\mathbf{x}_0)} \left[ w_{\boldsymbol{\phi}^*}^t(\mathbf{x}_t) \lambda(t) \left[ ||\mathbf{s}_{\boldsymbol{\theta}}(\mathbf{x}_t, t)||_2^2 - 2\nabla \log p_{\text{data}}(\mathbf{x}_t)^T \mathbf{s}_{\boldsymbol{\theta}}(\mathbf{x}_t, t) \right] \right] \mathrm{d}t + C_1 \tag{27}$$

$$= \frac{1}{2} \int_0^T \mathbb{E}_{p_{\text{bias}}(\mathbf{x}_0)} \mathbb{E}_{p_{0t}(\mathbf{x}_t|\mathbf{x}_0)} \left[ w_{\boldsymbol{\phi}^*}^t(\mathbf{x}_t) \lambda(t) \left[ ||\mathbf{s}_{\boldsymbol{\theta}}(\mathbf{x}_t, t)||_2^2 - 2\nabla \log p_{0t}(\mathbf{x}_t|\mathbf{x}_0)^T \mathbf{s}_{\boldsymbol{\theta}}(\mathbf{x}_t, t) \right] \right] \mathrm{d}t$$
$$- \frac{1}{2} \int_0^T \mathbb{E}_{p_{\text{bias}}(\mathbf{x}_0)} \mathbb{E}_{p_{0t}(\mathbf{x}_t|\mathbf{x}_0)} [2 w_{\boldsymbol{\phi}^*}^t(\mathbf{x}_t) \lambda(t) \nabla \log w_{\boldsymbol{\phi}^*}^t(\mathbf{x}_t)^T \mathbf{s}_{\boldsymbol{\theta}}(\mathbf{x}_t, t)] \mathrm{d}t + C_1 \tag{28}$$

$$= \frac{1}{2} \int_0^T \mathbb{E}_{p_{\text{bias}}(\mathbf{x}_0)} \mathbb{E}_{p_{0t}(\mathbf{x}_t|\mathbf{x}_0)} \left[ w_{\boldsymbol{\phi}^*}^t(\mathbf{x}_t) \lambda(t) \left[ ||\mathbf{s}_{\boldsymbol{\theta}}(\mathbf{x}_t, t) - \nabla \log p_{0t}(\mathbf{x}_t|\mathbf{x}_0)||_2^2 \right] \right] \mathrm{d}t + C_2$$
$$- \frac{1}{2} \int_0^T \mathbb{E}_{p_{\text{bias}}(\mathbf{x}_0)} \mathbb{E}_{p_{0t}(\mathbf{x}_t|\mathbf{x}_0)} [2 w_{\boldsymbol{\phi}^*}^t(\mathbf{x}_t) \lambda(t) \nabla \log w_{\boldsymbol{\phi}^*}^t(\mathbf{x}_t)^T \mathbf{s}_{\boldsymbol{\theta}}(\mathbf{x}_t, t)] \mathrm{d}t + C_1 \tag{29}$$

$$= \frac{1}{2} \int_0^T \mathbb{E}_{p_{\text{bias}}(\mathbf{x}_0)} \mathbb{E}_{p(\mathbf{x}_t|\mathbf{x}_0)} \left[ \lambda(t) w_{\boldsymbol{\phi}^*}^t(\mathbf{x}_t) \left[ ||\mathbf{s}_{\boldsymbol{\theta}}(\mathbf{x}_t, t) - \nabla \log p(\mathbf{x}_t|\mathbf{x}_0)||_2^2 \right. \right.$$
$$\left. \left. - 2\mathbf{s}_{\boldsymbol{\theta}}(\mathbf{x}_t, t)^T \nabla \log w_{\boldsymbol{\phi}^*}^t(\mathbf{x}_t) \right] \right] \mathrm{d}t + C_1 + C_2 \tag{30}$$

$$= \frac{1}{2} \int_0^T \mathbb{E}_{p_{\text{bias}}(\mathbf{x}_0)} \mathbb{E}_{p(\mathbf{x}_t|\mathbf{x}_0)} \left[ \lambda(t) w_{\boldsymbol{\phi}^*}^t(\mathbf{x}_t) \left[ ||\mathbf{s}_{\boldsymbol{\theta}}(\mathbf{x}_t, t) - \nabla \log p(\mathbf{x}_t|\mathbf{x}_0)||_2^2 \right. \right.$$
$$- 2\mathbf{s}_{\boldsymbol{\theta}}(\mathbf{x}_t, t)^T \nabla \log w_{\boldsymbol{\phi}^*}^t(\mathbf{x}_t) + 2\nabla \log p(\mathbf{x}_t|\mathbf{x}_0)^T \nabla \log w_{\boldsymbol{\phi}^*}^t(\mathbf{x}_t)$$
$$\left. \left. + ||\nabla \log w_{\boldsymbol{\phi}^*}^t(\mathbf{x}_t)||_2^2 \right] \right] \mathrm{d}t + C_1 + C_2 + C_3 \tag{31}$$

$$= \frac{1}{2} \int_0^T \mathbb{E}_{p_{\text{bias}}(\mathbf{x}_0)} \mathbb{E}_{p(\mathbf{x}_t|\mathbf{x}_0)} \left[ \lambda(t) w_{\boldsymbol{\phi}^*}^t(\mathbf{x}_t) \left[ ||\mathbf{s}_{\boldsymbol{\theta}}(\mathbf{x}_t, t) - \nabla \log p(\mathbf{x}_t|\mathbf{x}_0) - \nabla \log w_{\boldsymbol{\phi}^*}^t(\mathbf{x}_t)||_2^2 \right] \right] \mathrm{d}t + C \tag{32}$$

$$= \mathcal{L}_{\text{TIW-DSM}}(\boldsymbol{\theta}; p_{\text{bias}}, w_{\boldsymbol{\phi}^*}^t(\cdot)) + C \tag{33}$$

where $C_1, C_2, C_3, C$ be constants that do not depend on $\boldsymbol{\theta}$. Thus, $\mathcal{L}_{\text{TIW-DSM}}(\boldsymbol{\theta}; p_{\text{bias}}, w_{\boldsymbol{\phi}^*}^t(\cdot))$ is equivalent to $\mathcal{L}_{\text{SM}}(\boldsymbol{\theta}; p_{\text{data}})$ with respect to $\boldsymbol{\theta}$. $\qquad \square$

A.2  THEORETICAL ANALYSIS ON TIME-DEPENDENT DISCRIMINATOR TRAINING.

We further discuss the training objective of a time-dependent discriminator in eq. (34). We investigate whether optimizing at each time step of the density ratio would have a beneficial impact on other times. Theorem 3 provides some indirect answer. The minimization of log ratio estimation error at $t$ guarantees the smaller upper bound of the estimation error at $t = 0$ for a point.

$$\mathcal{L}_{\text{T-BCE}}(\phi; p_{\text{data}}, p_{\text{bias}}) := \int_0^T \lambda'(t) \left[ \mathbb{E}_{p_{\text{data}}^t(\mathbf{x}_t)}[\log d_\phi(\mathbf{x}_t, t)] + \mathbb{E}_{p_{\text{bias}}^t(\mathbf{x}_t)}[\log(1 - d_\phi(\mathbf{x}_t, t))] \right] \mathrm{d}t$$

(34)

**Theorem 3.** *Suppose the model density ratio $w_\phi^t$ and $\frac{p_{data}^t}{p_{bias}^t}$ are continuously differentiable on their supports with respect to $t$, for any $\mathbf{x}$. Assume $\frac{p_{data}^0}{p_{bias}^0}$ is nonzero at any $[0, 1]^d$, then we have*

$$\left| \log w_\phi^0(\mathbf{x}) - \log \frac{p_{data}^0(\mathbf{x})}{p_{bias}^0(\mathbf{x})} \right| \le \left| \log w_\phi^t(\mathbf{x}) - \log \frac{p_{data}^t(\mathbf{x})}{p_{bias}^t(\mathbf{x})} \right| + tC(\mathbf{x}, t; \phi) + O(t^2),$$

*where $C(\mathbf{x}, t; \phi) = \left| \frac{\partial}{\partial t} \log w_\phi^t(\mathbf{x}) - \frac{\partial}{\partial t} \log \frac{p_{data}^t(\mathbf{x})}{p_{bias}^t(\mathbf{x})} \right|$. For any $\epsilon > 0$, set $\phi_t^* = \arg\min_\phi \mathbb{E}_{[t,t+\epsilon)} \left[ \mathbb{E}_{p_{data}^t(\mathbf{x}_t)}[\log d_\phi(\mathbf{x}_t, t)] + \mathbb{E}_{p_{bias}^t(\mathbf{x}_t)}[\log(1 - d_\phi(\mathbf{x}_t, t))] \right]$. Then, the following properties hold:*

- $\log w_{\phi_t^*}^t(\mathbf{x}) = \log \frac{p_{data}^t(\mathbf{x})}{p_{bias}^t(\mathbf{x})}$.

- $C(\mathbf{x}, t; \phi_t^*) = 0$,

*for any $\mathbf{x}$. Therefore, at optimal $\phi_t^*$, the following inequality holds:*

$$\left| \log w_{\phi_t^*}^0(\mathbf{x}) - \log \frac{p_{data}^0(\mathbf{x})}{p_{bias}^0(\mathbf{x})} \right| \le O(t^2).$$

*Proof.* From the Taylor expansion with respect to $t$ variable, we have

$$\log w_\phi^0(\mathbf{x}) - \log \frac{p_{\text{data}}^0(\mathbf{x})}{p_{\text{bias}}^0(\mathbf{x})} = \log w_\phi^t(\mathbf{x}) - \log \frac{p_{\text{data}}^t(\mathbf{x})}{p_{\text{bias}}^t(\mathbf{x})}$$
$$+ t\left( \frac{\partial}{\partial t} \log w_\phi^t(\mathbf{x}) - \frac{\partial}{\partial t} \log \frac{p_{\text{data}}^t(\mathbf{x})}{p_{\text{bias}}^t(\mathbf{x})} \right) + O(t^2),$$

which derives

$$\left| \log w_\phi^0(\mathbf{x}) - \log \frac{p_{\text{data}}^0(\mathbf{x})}{p_{\text{bias}}^0(\mathbf{x})} \right| \le \left| \log w_\phi^t(\mathbf{x}) - \log \frac{p_{\text{data}}^t(\mathbf{x})}{p_{\text{bias}}^t(\mathbf{x})} \right| + tC(\mathbf{x}, t; \phi) + O(t^2),$$

by triangle inequality. Now, if $\phi_t^* = \arg\min_\phi \mathbb{E}_{[t,t+\epsilon)} \left[ \mathbb{E}_{p_{\text{data}}^t(\mathbf{x}_t)}[\log d_\phi(\mathbf{x}_t, t)] + \mathbb{E}_{p_{\text{bias}}^t(\mathbf{x}_t)}[\log(1 - d_\phi(\mathbf{x}_t, t))] \right]$, then $w_{\phi_t^*}^u(\mathbf{x}) = \frac{d_{\phi_t^*}(\mathbf{x}, u)}{1 - d_{\phi_t^*}(\mathbf{x}, u)} = \frac{p_{\text{data}}^u(\mathbf{x})}{p_{\text{bias}}^u(\mathbf{x})}$ for any $\mathbf{x}$ and $u \in [t, t+\epsilon)$. Therefore, we get

$$\left| \log w_{\phi_t^*}^t(\mathbf{x}) - \log \frac{p_{\text{data}}^t(\mathbf{x})}{p_{\text{bias}}^t(\mathbf{x})} \right| = 0$$

by plugging $t$ to $u$. Also, we get

$$\frac{\partial}{\partial t} \log w_{\phi_t^*}^t(\mathbf{x}) = \lim_{u \searrow t} \frac{\log w_{\phi_t^*}^u(\mathbf{x}) - \log w_{\phi_t^*}^t(\mathbf{x})}{u - t}$$
$$= \lim_{u \searrow t} \frac{\log \frac{p_{\text{data}}^u(\mathbf{x})}{p_{\text{bias}}^u(\mathbf{x})} - \log \frac{p_{\text{data}}^t(\mathbf{x})}{p_{\text{bias}}^t(\mathbf{x})}}{u - t}$$
$$= \frac{\partial}{\partial t} \log \frac{p_{\text{data}}^t(\mathbf{x})}{p_{\text{bias}}^t(\mathbf{x})},$$

since $\frac{p_{\text{data}}^t(\mathbf{x})}{p_{\text{bias}}^t(\mathbf{x})}$ is continuously differentiable with respect to $t$. Therefore, $C(\mathbf{x}, t; \phi_t^*) = 0$.  $\square$

### A.3 RELATION BETWEEN TIME-INDEPENDENT IMPORTANCE REWEIGHTING AND TIME-DEPENDENT IMPORTANCE REWEIGHTING

This section explains further equivalence between the objective functions of IW-DSM and TIW-DSM. We rewrite the objective function of time-independent importance reweighting as follows:

$$\mathcal{L}_{\text{IW-DSM}}(\boldsymbol{\theta}; p_{\text{bias}}, w_{\boldsymbol{\phi}^*}(\cdot)) \tag{35}$$

$$:= \frac{1}{2} \int_0^T \mathbb{E}_{p_{\text{bias}}(\mathbf{x}_0)} w_{\boldsymbol{\phi}^*}(\mathbf{x}_0) \mathbb{E}_{p(\mathbf{x}_t|\mathbf{x}_0)} \left[ \lambda(t) \big[ ||\mathbf{s}_{\boldsymbol{\theta}}(\mathbf{x}_t, t) - \nabla \log p(\mathbf{x}_t|\mathbf{x}_0)||_2^2 \big] \right] \mathrm{d}t,$$

where $w_{\boldsymbol{\phi}^*}(\mathbf{x}_0) := \frac{p_{\text{data}}(\mathbf{x}_0)}{p_{\text{bias}}(\mathbf{x}_0)}$. $\mathcal{L}_{\text{IW-DSM}}(\boldsymbol{\theta}; p_{\text{bias}}, w_{\boldsymbol{\phi}^*}(\cdot))$ is equivalent to $\mathcal{L}_{\text{DSM}}(\boldsymbol{\theta}; p_{\text{data}})$ as derived in eq. (6) and eq. (7). We also know the equivalence between $\mathcal{L}_{\text{TIW-DSM}}(\boldsymbol{\theta}; p_{\text{bias}}, w_{\boldsymbol{\phi}^*}^t(\cdot))$ and $\mathcal{L}_{\text{SM}}(\boldsymbol{\theta}; p_{\text{data}})$ from Theorem 1. Since $\mathcal{L}_{\text{SM}}(\boldsymbol{\theta}; p_{\text{data}})$ and $\mathcal{L}_{\text{DSM}}(\boldsymbol{\theta}; p_{\text{data}})$ are equivalent (Song & Ermon, 2019), we conclude that the objectives $\mathcal{L}_{\text{IW-DSM}}(\boldsymbol{\theta}; p_{\text{bias}}, w_{\boldsymbol{\phi}^*}(\cdot))$ and $\mathcal{L}_{\text{TIW-DSM}}(\boldsymbol{\theta}; p_{\text{bias}}, w_{\boldsymbol{\phi}^*}^t(\cdot))$ are equivalent w.r.t. $\boldsymbol{\theta}$ up to a constant.

This equivalence implies the empirical performance between IW-DSM and TIW-DSM is purely from the error propagation from the estimated time-independent density ratio $w_{\boldsymbol{\phi}}(\cdot)$ and the time-dependent density ratio $w_{\boldsymbol{\phi}}^t(\cdot)$.

### A.4 OBJECTIVE FOR INCORPORATING $\mathcal{D}_{\text{REF}}$

The objective functions of TIW-DSM and IW-DSM in the main paper explain how to treat $\mathcal{D}_{\text{bias}}$ for unbiased diffusion model training, but we actually have $\mathcal{D}_{\text{obs}} = \mathcal{D}_{\text{ref}} \cup \mathcal{D}_{\text{bias}}$. The objective that incorporates $\mathcal{D}_{\text{ref}}$ is necessary for better performance of the implementation.

To do this, we define the mixture distribution $p_{\text{obs}}^t := \frac{1}{2}p_{\text{bias}}^t + \frac{1}{2}p_{\text{data}}^t$, and plug $p_{\text{obs}}^t$ into $p_{\text{bias}}^t$ in each objective. Note that the density ratio between $p_{\text{data}}^t$ and $p_{\text{obs}}^t$ also can be represented by the time-dependent discriminator we explained in the main paper.

$$\begin{aligned}
\tilde{w}_{\boldsymbol{\phi}^*}^t(\mathbf{x}_t) &:= \frac{p_{\text{data}}^t(\mathbf{x}_t)}{p_{\text{obs}}^t(\mathbf{x}_t)} = \frac{p_{\text{data}}^t(\mathbf{x}_t)}{\frac{1}{2}p_{\text{bias}}^t(\mathbf{x}_t) + \frac{1}{2}p_{\text{data}}^t(\mathbf{x}_t)} \\
&= \frac{2\frac{p_{\text{data}}^t(\mathbf{x}_t)}{p_{\text{bias}}^t(\mathbf{x}_t)}}{1 + \frac{p_{\text{data}}^t(\mathbf{x}_t)}{p_{\text{bias}}^t(\mathbf{x}_t)}} = \frac{2w_{\boldsymbol{\phi}^*}^t(\mathbf{x}_t)}{1 + w_{\boldsymbol{\phi}^*}^t(\mathbf{x}_t)} = \frac{2\frac{d_{\boldsymbol{\phi}^*}(\mathbf{x}_t,t)}{1-d_{\boldsymbol{\phi}^*}(\mathbf{x}_t,t)}}{1 + \frac{d_{\boldsymbol{\phi}^*}(\mathbf{x}_t,t)}{1-d_{\boldsymbol{\phi}^*}(\mathbf{x}_t,t)}} = 2d_{\boldsymbol{\phi}^*}(\mathbf{x}_t, t) \tag{36}
\end{aligned}$$

By plugging $p_{\text{obs}}$ and $\tilde{w}_{\boldsymbol{\phi}^*}^t$ into our objective function, we can get the objective function that incorporates all the samples in $\mathcal{D}_{\text{obs}}$.

$$\mathcal{L}_{\text{TIW-DSM}}(\boldsymbol{\theta}; p_{\text{obs}}, \tilde{w}_{\boldsymbol{\phi}^*}^t(\cdot)) \tag{37}$$

$$:= \frac{1}{2} \int_0^T \mathbb{E}_{p_{\text{obs}}(\mathbf{x}_0)} \mathbb{E}_{p(\mathbf{x}_t|\mathbf{x}_0)} \left[ \lambda(t) \tilde{w}_{\boldsymbol{\phi}^*}^t(\mathbf{x}_t) \big[ ||\mathbf{s}_{\boldsymbol{\theta}}(\mathbf{x}_t, t) - \nabla \log p(\mathbf{x}_t|\mathbf{x}_0) - \nabla \log \tilde{w}_{\boldsymbol{\phi}^*}^t(\mathbf{x}_t)||_2^2 \big] \right] \mathrm{d}t$$

In the same spirit, the time-independent importance reweighting objective that incorporates $\mathcal{D}_{\text{ref}}$ represented as follows:

$$\mathcal{L}_{\text{IW-DSM}}(\boldsymbol{\theta}; p_{\text{obs}}, \tilde{w}_{\boldsymbol{\phi}^*}(\cdot)) \tag{38}$$

$$:= \frac{1}{2} \int_0^T \mathbb{E}_{p_{\text{obs}}(\mathbf{x}_0)} \tilde{w}_{\boldsymbol{\phi}^*}(\mathbf{x}_0) \mathbb{E}_{p(\mathbf{x}_t|\mathbf{x}_0)} \left[ \lambda(t) \big[ ||\mathbf{s}_{\boldsymbol{\theta}}(\mathbf{x}_t, t) - \nabla \log p(\mathbf{x}_t|\mathbf{x}_0)||_2^2 \big] \right] \mathrm{d}t,$$

where $\tilde{w}_{\boldsymbol{\phi}^*}(\mathbf{x}_0) = \frac{p_{\text{data}}^0(\mathbf{x}_0)}{p_{\text{obs}}^0(\mathbf{x}_0)}$.

The DSM(obs) in our experiment optimize the following objective in eq. (39).

$$\mathcal{L}_{\text{DSM}}(\boldsymbol{\theta}; p_{\text{obs}}) = \frac{1}{2} \int_0^T \mathbb{E}_{p_{\text{obs}}(\mathbf{x}_0)} \mathbb{E}_{p(\mathbf{x}_t|\mathbf{x}_0)} \left[ \lambda(t) \big[ ||\mathbf{s}_{\boldsymbol{\theta}}(\mathbf{x}_t, t) - \nabla \log p(\mathbf{x}_t|\mathbf{x}_0)||_2^2 \big] \right] \mathrm{d}t \tag{39}$$

## A.5 Loss component ablations

The proposed objective function, eq. (40), utilizes $w_{\phi^*}$ as two roles in our method: 1) reweighting and 2) score correction. We discuss what if each component did not exist.

$$\mathcal{L}_{\text{TIW-DSM}}(\boldsymbol{\theta}; p_{\text{bias}}, w_{\phi^*}^t(\cdot)) \tag{40}$$

$$:= \frac{1}{2} \int_0^T \mathbb{E}_{p_{\text{bias}}(\mathbf{x}_0)} \mathbb{E}_{p(\mathbf{x}_t|\mathbf{x}_0)} \left[ \lambda(t) w_{\phi^*}^t(\mathbf{x}_t) \left[ ||\mathbf{s}_{\boldsymbol{\theta}}(\mathbf{x}_t, t) - \nabla \log p(\mathbf{x}_t|\mathbf{x}_0) - \nabla \log w_{\phi^*}^t(\mathbf{x}_t)||_2^2 \right) \right] \right] dt$$

First, we consider the objective function that only takes the score correction:

$$\frac{1}{2} \int_0^T \mathbb{E}_{p_{\text{bias}}(\mathbf{x}_0)} \mathbb{E}_{p(\mathbf{x}_t|\mathbf{x}_0)} \left[ \lambda(t) \left[ ||\mathbf{s}_{\boldsymbol{\theta}}(\mathbf{x}_t, t) - \nabla \log p(\mathbf{x}_t|\mathbf{x}_0) - \nabla \log w_{\phi^*}^t(\mathbf{x}_t)||_2^2 \right) \right] \right] dt. \tag{41}$$

If we define newly parameterized distribution $\mathbf{s}_{\boldsymbol{\theta}}'(\mathbf{x}_t, t) := \mathbf{s}_{\boldsymbol{\theta}}(\mathbf{x}_t, t) - \nabla \log w_{\phi^*}^t(\mathbf{x}_t)$ as the model distribution (the adjusting parameter is still only $\boldsymbol{\theta}$), the objective becomes like eq. (42).

$$\frac{1}{2} \int_0^T \mathbb{E}_{p_{\text{bias}}(\mathbf{x}_0)} \mathbb{E}_{p(\mathbf{x}_t|\mathbf{x}_0)} \left[ \lambda(t) \left[ ||s_{\boldsymbol{\theta}}'(\mathbf{x}_t, t) - \nabla \log p(\mathbf{x}_t|\mathbf{x}_0)||_2^2 \right) \right] \right] dt \tag{42}$$

This objective is same as $\mathcal{L}_{\text{DSM}}$ with $p_{\text{bias}}$, so $s_{\boldsymbol{\theta}}'(\mathbf{x}_t, t)$ will converge to $\nabla \log p_{\text{bias}}(\mathbf{x}_t)$. By the relation from $\mathbf{s}_{\boldsymbol{\theta}}(\mathbf{x}_t, t) = s_{\boldsymbol{\theta}}'(\mathbf{x}_t, t) + \nabla \log w_{\phi^*}^t(\mathbf{x}_t)$, $\mathbf{s}_{\boldsymbol{\theta}}(\mathbf{x}_t, t)$ will converges to $\nabla \log p_{\text{bias}}(\mathbf{x}_t) + \nabla \log \frac{p_{\text{data}}(\mathbf{x}_t)}{p_{\text{bias}}(\mathbf{x}_t)} = \nabla \log p_{\text{data}}(\mathbf{x}_t)$. This means that only applying score correction guarantees optimality, so this is the reason for the quite good performance.

Second, we consider the objective function that only takes the time-dependent reweighting:

$$\frac{1}{2} \int_0^T \mathbb{E}_{p_{\text{bias}}(\mathbf{x}_0)} \mathbb{E}_{p(\mathbf{x}_t|\mathbf{x}_0)} \left[ \lambda(t) w_{\phi^*}^t(\mathbf{x}_t) \left[ ||\mathbf{s}_{\boldsymbol{\theta}}(\mathbf{x}_t, t) - \nabla \log p(\mathbf{x}_t|\mathbf{x}_0)||_2^2 \right) \right] \right] dt. \tag{43}$$

We can derive that eq. (43) is equivalent to following objective in eq. (44).

$$\frac{1}{2} \int_0^T \mathbb{E}_{p_{\text{bias}}^t(\mathbf{x}_t)} \left[ \lambda(t) w_{\phi^*}^t(\mathbf{x}_t) \left[ ||\mathbf{s}_{\boldsymbol{\theta}}(\mathbf{x}_t, t) - \nabla \log p_{\text{bias}}(\mathbf{x}_t)||_2^2 \right) \right] \right] dt, \tag{44}$$

which implies that $\mathbf{s}_{\boldsymbol{\theta}}(\mathbf{x}_t, t)$ will converge to $\nabla \log p_{\text{bias}}(\mathbf{x}_t)$. This is the reason that the objective without score correction performs similarly to DSM(obs).

## A.6 Generalized objective function by adjusting density ratio

We generalize our objective for the ablation study in Section 4.3 by adjusting the density ratio, which is represented as eq. (45).

$$\mathcal{L}_{\text{TIW-DSM}}(\boldsymbol{\theta}; p_{\text{bias}}, w_{\phi^*}^t(\cdot)^\alpha) \tag{45}$$

$$:= \frac{1}{2} \int_0^T \mathbb{E}_{p_{\text{bias}}(\mathbf{x}_0)} \mathbb{E}_{p(\mathbf{x}_t|\mathbf{x}_0)} \left[ \lambda(t) w_{\phi^*}^t(\mathbf{x}_t)^\alpha \left[ ||\mathbf{s}_{\boldsymbol{\theta}}(\mathbf{x}_t, t) - \nabla \log p(\mathbf{x}_t|\mathbf{x}_0) - \alpha \nabla \log w_{\phi^*}^t(\mathbf{x}_t)||_2^2 \right) \right] \right] dt$$

As $\alpha \to 0$, $\mathcal{L}_{\text{TIW-DSM}}(\boldsymbol{\theta}; p_{\text{bias}}, w_{\phi^*}^t(\cdot)^\alpha)$ becomes $\mathcal{L}_{\text{DSM}}(\boldsymbol{\theta}; p_{\text{bias}})$, i.e.,

$$\mathcal{L}_{\text{TIW-DSM}}(\boldsymbol{\theta}; p_{\text{bias}}, w_{\phi^*}^t(\cdot)^0) \tag{46}$$

$$= \frac{1}{2} \int_0^T \mathbb{E}_{p_{\text{bias}}(\mathbf{x}_0)} \mathbb{E}_{p(\mathbf{x}_t|\mathbf{x}_0)} \left[ \lambda(t) \left[ ||\mathbf{s}_{\boldsymbol{\theta}}(\mathbf{x}_t, t) - \nabla \log p(\mathbf{x}_t|\mathbf{x}_0)||_2^2 \right) \right] \right] dt = \mathcal{L}_{\text{DSM}}(\boldsymbol{\theta}; p_{\text{bias}})$$

To adopt this scaling to our objective with incorporate $\mathcal{D}_{\text{ref}}$, we utilize the relation in eq. (47), and define $\tilde{w}_{\phi^*}^t(\mathbf{x}_t, \alpha)$ through eq. (48).

$$\tilde{w}_{\phi^*}^t(\mathbf{x}_t) = \frac{2 w_{\phi}^t(\mathbf{x}_t)}{1 + w_{\phi^*}^t(\mathbf{x}_t)} \tag{47}$$

$$\tilde{w}_{\phi*}^t(\mathbf{x}_t, \alpha) := \frac{2w_\phi^t(\mathbf{x}_t)^\alpha}{1 + w_{\phi*}^t(\mathbf{x}_t)^\alpha} \tag{48}$$

Then, the $\alpha$-generalized objective that incorporates $\mathcal{D}_{\text{ref}}$ can be expressed as follows.

$$\mathcal{L}_{\text{TIW-DSM}}(\boldsymbol{\theta}; p_{\text{obs}}, \tilde{w}_{\phi*}^t(\cdot, \alpha)) \tag{49}$$

$$:= \frac{1}{2} \int_0^T \mathbb{E}_{p_{\text{obs}}(\mathbf{x}_0)} \mathbb{E}_{p(\mathbf{x}_t|\mathbf{x}_0)} \left[ \lambda(t) \tilde{w}_{\phi*}^t(\mathbf{x}_t, \alpha) \left[ \|\mathbf{s}_{\boldsymbol{\theta}}(\mathbf{x}_t, t) - \nabla \log p(\mathbf{x}_t|\mathbf{x}_0) - \nabla \log \tilde{w}_{\phi*}^t(\mathbf{x}_t, \alpha)\|_2^2) \right] \right] \mathrm{d}t$$

As $\alpha \to 0$, $\tilde{w}_{\phi*}^t(\mathbf{x}_t, \alpha)$ becomes 1, which leads $\nabla \log \tilde{w}_{\phi*}^t(\mathbf{x}_t, \alpha)$ be 0.

$$\mathcal{L}_{\text{TIW-DSM}}(\boldsymbol{\theta}; p_{\text{obs}}, \tilde{w}_{\phi*}^t(\cdot, 0))$$

$$= \frac{1}{2} \int_0^T \mathbb{E}_{p_{\text{obs}}(\mathbf{x}_0)} \mathbb{E}_{p(\mathbf{x}_t|\mathbf{x}_0)} \left[ \lambda(t) \tilde{w}_{\phi*}^t(\mathbf{x}_t, 0) \left[ \|\mathbf{s}_{\boldsymbol{\theta}}(\mathbf{x}_t, t) - \nabla \log p(\mathbf{x}_t|\mathbf{x}_0) - \nabla \log \tilde{w}_{\phi*}^t(\mathbf{x}_t, 0)\|_2^2) \right] \right] \mathrm{d}t$$

$$= \frac{1}{2} \int_0^T \mathbb{E}_{p_{\text{obs}}(\mathbf{x}_0)} \mathbb{E}_{p(\mathbf{x}_t|\mathbf{x}_0)} \left[ \lambda(t) \left[ \|\mathbf{s}_{\boldsymbol{\theta}}(\mathbf{x}_t, t) - \nabla \log p(\mathbf{x}_t|\mathbf{x}_0)\|_2^2) \right] \right] \mathrm{d}t$$

$$= \mathcal{L}_{\text{DSM}}(\boldsymbol{\theta}; p_{\text{obs}}) \tag{50}$$

This implies that $\alpha$ interpolates the objective function between DSM(obs) and TIW-DSM. We can observe the quantitative results also interpolated in the range of $\alpha \in [0, 1]$ as shown in Figure 7.

## B  RELATED WORK

### B.1  FAIRNESS IN ML & GENERATIVE MODELING

Fairness is widely studied in the fields of classification tasks (Dwork et al., 2012; Feldman et al., 2015; Heidari et al., 2018; Adel et al., 2019), representation learning (Zemel et al., 2013; Louizos et al., 2015; Song et al., 2019), and generative modeling (Um & Suh, 2023; Sattigeri et al., 2019; Xu et al., 2018; Teo et al., 2023). In terms of classification tasks, the objective for fairness is mainly to handle a classifier to be independent of the sensitive attributes such as gender with different measurement metrics (Hardt et al., 2016; Feldman et al., 2015). Fair representation learning is defined as equal representation which is a uniform distribution of samples with respect to the sensitive attributes (Hutchinson & Mitchell, 2019).

The task we address in this paper is also called *fair* generative modeling (Xu et al., 2018; Choi et al., 2020; Teo et al., 2023), which aims to estimate a balanced distribution of samples with respect to sensitive attributes. With regard to data generation, there are relevant works such as Fair-GAN (Xu et al., 2018) and FairnessGAN (Sattigeri et al., 2019). These methods have been advanced to generate data instances characterized by fairness attributes, with their respective labels. These generated data instances are utilized as a preprocessing step. On the other hand, Teo et al. (2023) introduces transfer learning to learn a fair generative model. They adapt the pre-trained generative model trained by large, biased datasets via leveraging the small, unbiased reference dataset to finefune the model. Choi et al. (2020); Um & Suh (2023) treat fair generative modeling under a weak supervision setting so utilize the small amount of reference dataset. Most of the fair generative models have progressed using GANs. In the diffusion models, we propose, that the concept relevant to fairness & dataset bias has not yet received significant attention.

Friedrich et al. (2023) is a concurrent study that explores the theme of fairness in diffusion models, but their work is distinctly differentiated from our paper in terms of problem setting and methodology. Our paper focuses on a weak supervision setting, which is a cost-effective scenario in terms of dataset collection. Conversely, Friedrich et al. (2023) leverage information in the joint space of (text, image) using a pre-trained text conditional diffusion model. This implies that their approach relies on point-wise text supervision to mitigate bias. There is also a distinguishable difference in the methods. Friedrich et al. (2023) is based on the guidance method, which requires 2 to 3 times more NFEs for sampling. Our paper proposes the objective function for unbiased score network training, so we only need 1 NFE of score network at every denoising step. Please refer to Appendix E.6 for quantitative comparison.

### B.2  IMPORTANCE REWEIGHTING

There are many approaches to reweighting data points for their purpose, which is common in the fields of noisy label learning (Liu & Tao, 2015; Wang et al., 2017a), class imbalanced learning (Ren et al., 2018; Guo et al., 2022; Duggal et al., 2021; Park et al., 2021), and fairness (Chai & Wang, 2022; Hu et al., 2023; Krasanakis et al., 2018; Iosifidis & Ntoutsi, 2019). In the context of learning with noisy labels, importance reweighting aims to adjust the loss function by assigning reduced weights to instances with noisy labels and elevated weights to instances with clean labels, thereby mitigating the impact of noisy labels on the learning process Liu & Tao (2015). Similar to the concept from the noisy label, research from class imbalanced learning utilizes an importance reweighting scheme to prevent the model from being biased to the majority classes while amplifying the effects of minority classes (Wang et al., 2017b; Ren et al., 2018; Guo et al., 2022). In terms of fairness, there are researches for importance reweighting (Chai & Wang, 2022; Hu et al., 2023). These works on fairness aim to mitigate representation bias which is caused by insufficient and imbalanced data instances in a fair perspective. Consequently, they propose instance reweighting as a means to facilitate fair representation learning within the model.

The reweighting related to time $t$ is considered in diffusion models (Nichol & Dhariwal, 2021; Song et al., 2021; Kim et al., 2022a). However, these studies focus on resampling and reweighting the random variable $t$ itself, while we focus on the reweighting $\mathbf{x}_t$.

### B.3 Score correction in diffusion model

The sampling process of the diffusion model involves an iterative update process using a score direction, typically approximated by the score network. When there is a specific purpose for generating data, score correction becomes necessary. There are several methods to adjust this score direction, each tailored to specific purposes. From a technical standpoint, these methods can be divided into two groups: guidance methods and score-matching regularization methods.

Guidance methods introduce additional gradient signals to adjust the update direction. Classifier guidance (Song et al., 2020; Dhariwal & Nichol, 2021) utilizes a gradient signal from a classifier to generate samples that satisfy a condition. Classifier-free guidance (Ho & Salimans, 2021) also aims at conditional generation but relies on both unconditional and conditional scores. Furthermore, various methods have been proposed to enable controllable generation using auxiliary models with a pre-trained unconditional score model (Graikos et al., 2022; Song et al., 2023). On the other hand, discriminator guidance (Kim et al., 2023) serves a different purpose by enhancing the sampling performance of a diffusion model through the use of a discriminator that distinguishes between real images and generated images. EGSDE (Zhao et al., 2022) leverages guidance signals based on energy functions, enhancing unpaired image-to-image translation. Guidance methods have the advantage of utilizing pre-trained score networks without the need for additional training. However, they require separate network training for guidance and additional network evaluation during the sampling process.

There is a body of work on score-matching regularization for better likelihood estimation (Lu et al., 2022; Zheng et al., 2023b; Lai et al., 2023). Na et al. (2024) propose a regularized conditional score-matching objective to mitigate label noise. The unique benefit of score-matching regularization is that it does not require an additional network at the inference stage.

### B.4 Time-dependent density ratio in GANs

Density ratio is closely associated with the training of GANs (Goodfellow et al., 2014; Nowozin et al., 2016; Uehara et al., 2016). The discrimination between perturbed real data and perturbed generated data is often mentioned in GAN literature. This is because the discriminator of a GAN also suffers from a density-chasm problem, and a noise injection trick could resolve it. Arjovsky & Bottou (2017) propose a method to perturb real data and fake data with a small gaussian noise scale for discriminator input, but the practical choice of noise scale in high-dimension is not easy (Roth et al., 2017). Wang et al. (2023b) propose a multi-scale noise injection using a forward diffusion process and introduced an adaptive diffusion technique, achieving significant performance improvements in high-dimensional datasets. Xiao et al. (2022); Zheng et al. (2023a) utilize GAN's generator to achieve fast sampling in the reverse diffusion process, and they also naturally conduct discrimination between perturbed distribution. However, the time-dependent discriminator in GANs fundamentally differs in its use case from the proposed method that serves the roles of reweighting and score correction.

## C Overfitting with limited data

We observe the FID overfitting phenomenon when we train diffusion models with too small a subset of data. In GANs, the origin of overfitting is well elucidated by Karras et al. (2020). However, in diffusion models, the origin of overfitting is not well explored but often reported from the literature (Nichol & Dhariwal, 2021; Moon et al., 2022; Song & Ermon, 2020). Training configurations, such as network architecture, EMA, and diffusion noise scheduling, affect this phenomenon. One thing explicitly observed from Figure 10 is that overfitting becomes serious when the number of data becomes smaller. Our experiment sometimes considers a small amount of data, so we periodically measure the FID and choose the best one.

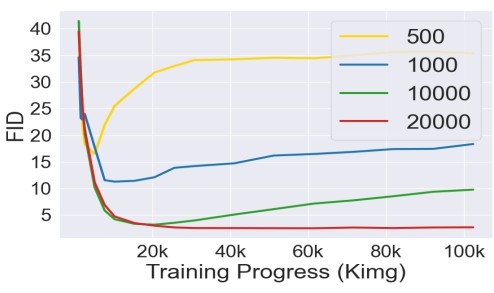 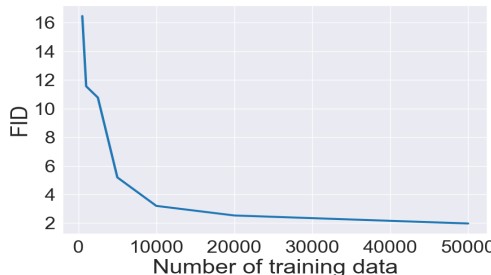

(a) Training curve on various numbers of data          (b) FID under various # of data with early stopping

Figure 10: Overfitting in FID with a limited number of training data in training diffusion model with CIFAR-10.

# D   IMPLEMENTATION DETAIL

## D.1   DATSETS

We explain the details of the dataset construction for our experiment. Table 5 shows the information about $\mathcal{D}_{\text{bias}}$, $\mathcal{D}_{\text{ref}}$ and the entire unbiased dataset. To construct $\mathcal{D}_{\text{bias}}$, we define the bias statistics in each latent subgroup (See Figure 11 for the proportion), and we randomly sampled from each subgroup. Once we established $\mathcal{D}_{\text{bias}}$, we conducted experiments using the same set for all baselines. The ground truth bias information on each data point is provided from the official dataset in CIFAR-10, CIFAR-100, and CelebA. We use bias information for FFHQ from https://github.com/DCGM/ffhq-features-dataset. The entire unbiased dataset is used to construct $\mathcal{D}_{\text{ref}}$ and evaluation. We set the entire unbiased dataset as almost the maximum number of samples that are balanced under latent statistics. The reference dataset $\mathcal{D}_{\text{ref}}$ is randomly sampled from the entire unbiased dataset. Note that we do not intentionally balance the latent statistics in $\mathcal{D}_{\text{ref}}$, and we use the same $\mathcal{D}_{\text{ref}}$ for all baselines.

Table 5: Dataset configurations

|  | CIFAR-10 | CIFAR-100 | FFHQ | CelebA |
|---|---|---|---|---|
| **Resolution** | $3 \times 32 \times 32$ | $3 \times 32 \times 32$ | $3 \times 64 \times 64$ | $3 \times 64 \times 64$ |
| **Bias dataset** $\mathcal{D}_{\text{bias}}$ |  |  |  |  |
| Number of instances | 10000 | 10000 | 40000 | 162770 |
| Bias factor | Class | Class | Gender | (Gender, Hair color) |
| Bias subgroup | 10 | 100 | 2 | (2, 2) |
| Bias type | Long tail | Long tail | 80%, 90% | Benchmark |
| **Entire unbiased dataset** |  |  |  |  |
| Number of instances | 50000 | 50000 | 50000 | 75136 |
| Number of instances in each bias group | 5000 | 500 | 25000 | 18784 |
| **Reference dataset** $\mathcal{D}_{\text{ref}}$ |  |  |  |  |
| Number of instances | 500, 1000, 2500, 5000 | 500, 1000, 2500, 5000 | 500, 5000 | 8140 |

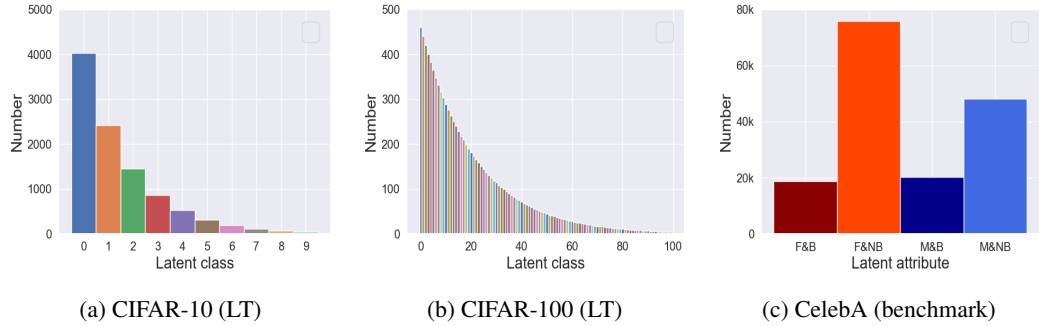

(a) CIFAR-10 (LT)  (b) CIFAR-100 (LT)  (c) CelebA (benchmark)

Figure 11: The latent statistics in each $\mathcal{D}_{\text{bias}}$.

## D.2 TRAINING CONFIGURATION

We follow the procedures outlined in EDM (Karras et al., 2022) to implement the diffusion models only by changing the learning batch size and objective functions. For the time-dependent discriminator, we follow DG (Kim et al., 2023). Table 6 presents the details of our experiment. We utilize the model architecture, and training configuration of diffusion model from `https://github.com/NVlabs/edm`. For CIFAR-10 and CIFAR-100 experiments, we follow the best setting that is used for CIFAR-10 in EDM. For FFHQ and CelebA experiments, we follow the best setting that is used for FFHQ in EDM, except for batch size. For time-dependent discriminator, we utilize the setting from `https://github.com/alsdudrla10/DG`. The time-dependent discriminator consists of two U-Net encoder architectures. We use a pre-trained U-Net encoder from ADM `https://github.com/openai/guided-diffusion` which is as a feature extractor. We train the shallow U-Net encoder that transforms from the feature to the logit. For sampling, we utilize EDM deterministic sampler. To implement a time-independent discriminator for IW-DSM, we utilize the same discriminator architecture but only feed-forward $t = 0$ for time inputs.

Table 6: Training and sampling configurations.

|  | CIFAR-10 | CIFAR-100 | FFHQ | CelebA |
|---|---|---|---|---|
| **Score Network Architecture** | | | | |
| Backbone U-net | DDPM++ | DDPM++ | DDPM++ | DDPM++ |
| Channel multiplier | 128 | 128 | 128 | 128 |
| Channel per resolution | 2-2-2 | 2-2-2 | 1-2-2-2 | 1-2-2-2 |
| **Score Network Training** | | | | |
| Learning rate $\times 10^4$ | 10 | 10 | 2 | 2 |
| Augment probability | 12% | 12% | 15% | 15% |
| Dropout probability | 13% | 13% | 5% | 5% |
| Batch size | 256 | 256 | 128 | 128 |
| **Discriminator Architecture** | | | | |
| Feature extractor | ADM | ADM | ADM | ADM |
| Backbone | U-Net encoder | U-Net encoder | U-Net encoder | U-Net encoder |
| depth | 2 | 2 | 2 | 2 |
| width | 128 | 128 | 128 | 128 |
| Attention Resolutions | 32,16, 8 | 32,16, 8 | 32,16, 8 | 32,16, 8 |
| Model channel | 128 | 128 | 128 | 128 |
| **Discriminator Training** | | | | |
| Batch size | 128 | 128 | 128 | 128 |
| Perturbation | VP | VP | Cosine VP | Cosine VP |
| Time sampling | Importance | Importance | Importance | Importance |
| Learning rate $\times 10^3$ | 4 | 4 | 4 | 4 |
| Iteration | 10k | 10k | 10k | 10k |
| **Sampling** | | | | |
| Solver type | ODE | ODE | ODE | ODE |
| Solver order | 2 | 2 | 2 | 2 |
| NFE | 35 | 35 | 79 | 79 |

### D.3 METRIC

FID measures the distance between the sample distributions. Each group of samples is projected into the pre-trained features space and approximated through Gaussian distribution. So, FID measures both sample fidelity and diversity. We consider this to be the metric to indicate how well the model distribution approximates an unbiased data distribution. We utilize `https://github.com/NVlabs/edm` for FID computation.

For the analysis purpose, we use the metrics recall. The recall describes how well the generated samples in the feature space cover the manifold of unbiased data. We utilize this metric to highlight the reason why IW-DSM shows so poor FID performance in . We utilize `https://github.com/chen-hao-chao/dlsm` for recall computation.

*Bias* (Choi et al., 2020) is also utilized for the analysis. This metric measures how similar latent statistics are to the reference data. This metric requires a pre-trained classifier $p_\psi$ that distinguishes the latent subgroups. The classifier trained on the entire unbiased dataset. We use a pre-trained *vgg13-bn* model from `https://github.com/huyvnphan/PyTorch_CIFAR10` for CIFAR-10, pre-trained *DenseNet-BC (L=190, k=40)* from `https://github.com/bearpaw/pytorch-classification` for CIFAR-100. This latent classifier is also used to compute the portion of the latent group for sample visualization. For FFHQ and CelebA, we utilize our discriminator architecture with only feed-forward $t = 0$, and adjust the output channels.

$$Bias := \Sigma_z ||\mathbb{E}_{\mathbf{x}\sim\mathcal{D}_{\mathrm{ref}}}[p(z|\mathbf{x})] - \mathbb{E}_{\mathbf{x}\sim p_{\boldsymbol{\theta}}}[p(z|\mathbf{x})]||_2 \tag{51}$$

### D.4 ALGORITHM

---

**Algorithm 1:** Discriminator Training algorithm

    **Input:** Reference data $\mathcal{D}_{\mathrm{ref}}$, biased data $\mathcal{D}_{\mathrm{bias}}$, perturbation kernel $p_{t|0}$, temporal weights $\lambda$
    **Output:** Discriminator $d_{\boldsymbol{\phi}}$
1  **while** *not converged* **do**
2     |  Sample $\mathbf{x}_1, ..., \mathbf{x}_{B/2}$ from $\mathcal{D}_{\mathrm{ref}}$
3     |  Sample $\mathbf{x}_{B/2+1}, ..., \mathbf{x}_B$ from $\mathcal{D}_{\mathrm{bias}}$
4     |  Sample time $t_1, ..., t_{B/2}, t_{B/2+1}, ..., t_B$ from $[0, T]$
5     |  Diffuse $\mathbf{x}_1^{t_1}, ..., \mathbf{x}_{B/2}^{t}, \mathbf{x}_{B/2+1}^{t}, ..., \mathbf{x}_B^{t_B}$ using the transition kernel $p_{t|0}$
6     |  $l \leftarrow -\sum_{i=1}^{B/2} \lambda(t_i) \log d_{\boldsymbol{\phi}}(\mathbf{x}_i, t_i) - \sum_{i=B/2+1}^{B} \lambda(t_i) \log(1 - d_{\boldsymbol{\phi}}(\mathbf{x}_i, t_i))$
7     |  Update $\boldsymbol{\phi}$ by $l$ using the gradient descent method
8  **end**

---

**Algorithm 2:** Score Training algorithm with TIW-DSM

    **Input:** Observed data $\mathcal{D}_{\mathrm{obs}}$, discriminator $\phi^*$, perturbation kernel $p_{t|0}$, temporal weights $\lambda$
    **Output:** Score network $\mathbf{s}_{\boldsymbol{\theta}}$
1  **while** *not converged* **do**
2     |  Sample $\mathbf{x}_0$ from $\mathcal{D}_{obs}$, and time $t$ from $[0, T]$
3     |  Sample $\mathbf{x}_t$ from the transition kernel $p_{t|0}$
4     |  Evaluate $\tilde{w}_{\phi^*}^t(\mathbf{x}_t)$ using eq. (36)
5     |  $l \leftarrow \lambda(t)\tilde{w}_{\phi^*}^t(\mathbf{x}_t)||\mathbf{s}_{\boldsymbol{\theta}}(\mathbf{x}_t, t) - \nabla \log p(\mathbf{x}_t|\mathbf{x}_0) - \nabla \log \tilde{w}_{\phi^*}^t(\mathbf{x}_t)||_2^2$
6     |  Update $\boldsymbol{\theta}$ by $l$ using the gradient descent method
7  **end**

---

## D.5 COMPUTATIONAL COST

In this section, we compare the TIW-DSM and IW-DSM regarding computational costs. Both methods require the evaluation of the discriminator during the training phase, but the evaluation procedures are somewhat different. IW-DSM only requires the feed-forward value of the discriminator. On the other hand, TIW-DSM requires the value $\nabla \log w_{\phi^*}^t(\cdot)$, which necessitates auto gradient operation in PyTorch. This slightly increases both training time and memory usage. Once the training is complete, the discriminator is not used for sampling, so the sampling time and memory remain the same. Table 7 shows the computational costs measured using RTX 4090 $\times$ 4 cores in the CIFAR-10 experiments. Note that the training time-dependent discriminator is negligibly cheap, converging around 10 minutes with 1 RTX 4090.

Table 7: The computational cost comparison between IW-DSM and TIW-DSM.

|  | IW-DSM | TIW-DSM |
|---|---|---|
| Training time | 0.26 Second / Batch | 0.34 Second / Batch |
| Training memory | 13,258 MiB $\times$ 4 Core | 15,031 MiB $\times$ 4 Core |
| Sampling time | 7.5 Minute / 50k | 7.5 Minute / 50k |
| Sampling memory | 4,928 MiB $\times$ 4 Core | 4,928 MiB $\times$ 4 Core |

## E ADDITIONAL EXPERIMENTAL RESULT

### E.1 COMPARISON TO GAN BASELINES

The reason we developed a methodology with a focus on the diffusion model is because it demonstrates superior sample quality compared to other generative models like GANs. To validate this, we conducted experiments with a GAN baselines. Table 8 compare the performance with GAN. GAN(ref) and GAN(obs) indicates the GAN training with $\mathcal{D}_{ref}$ and $\mathcal{D}_{obs}$, repectively. IW-GAN is applying importance reweighting on GAN training (Choi et al., 2020). We observed training GAN with limited data resulted in failure, which is often discussed in the literatures (Karras et al., 2020). IW-GAN also exhibited a similar phenomenon, as it hardly utilized the information from $\mathcal{D}_{bias}$, as discussed in Section 4.4. The issue with limited data actually led to better performance when we used all the observed data. Due to these issues, the quantitative metrics did not perform well, so we removed them from our considerations. We utilize the code from `https://github.com/ermongroup/fairgen` and modify the resolution for CIFAR-10. Figure 12 shows the samples from GANs.

Table 8: Comparision to GAN baselines on CIFAR-10 (LT) experiment. The reported value is FID ($\downarrow$).

|  | Reference size | | | |
|---|---|---|---|---|
|  | 5% | 10% | 25% | 50% |
| GAN(ref) | 284.11 | 246.75 | 144.32 | 56.29 |
| GAN(obs) | 42.09 | 36.45 | 35.67 | 34.42 |
| IW-GAN | 260.32 | 235.22 | 120.23 | 50.32 |
| IW-DSM | 15.79 | 11.45 | 8.19 | 4.28 |
| TIW-DSM | **11.51** | **8.08** | **5.59** | **4.06** |

### E.2 TRAINIG CURVE

We provide more training curves on CIFAR-10 and CIFAR-100 experiments. We measured the FID in increments of 2.5K images during the early stages of training and then in increments of 10K images after reaching 20K images for all our experiments. See Figure 13 for training curves, which demonstrate the training stability of TIW-DSM.

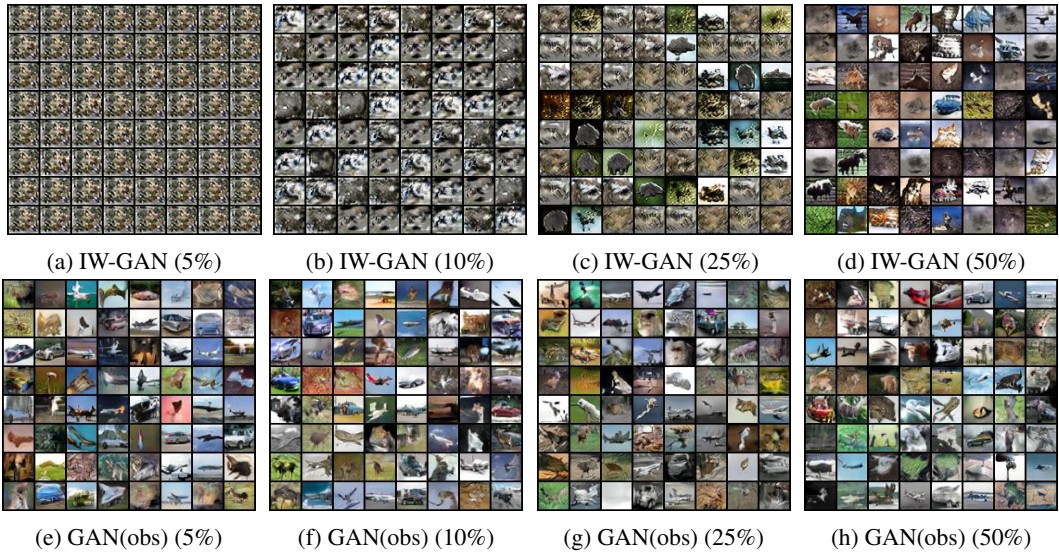

(a) IW-GAN (5%)  (b) IW-GAN (10%)  (c) IW-GAN (25%)  (d) IW-GAN (50%)

(e) GAN(obs) (5%)  (f) GAN(obs) (10%)  (g) GAN(obs) (25%)  (h) GAN(obs) (50%)

Figure 12: Samples from GAN baselines according to the method and reference sizes.

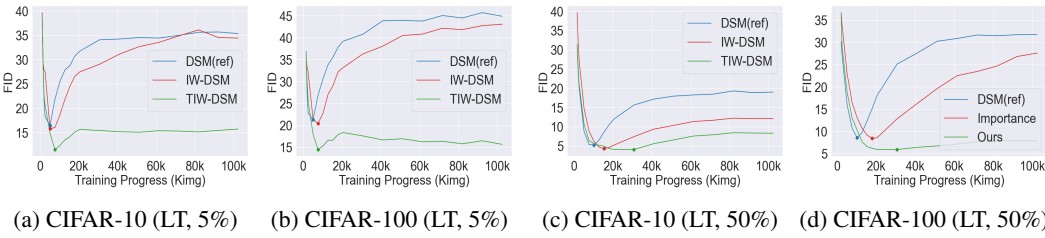

(a) CIFAR-10 (LT, 5%)  (b) CIFAR-100 (LT, 5%)  (c) CIFAR-10 (LT, 50%)  (d) CIFAR-100 (LT, 50%)

Figure 13: Training curves on CIFAR-10 / CIFAR-100 experiments.

### E.3 SAMPLE COMPARISON

We further provide the samples from each experiment in Figures 15 to 18. We examine the proportion of each latent group on samples through Appendix D.3, and reflect the latent statistics on each generated sample. Figure 14 shows more examples that the conversion from majority group to minority group through the proposed method in CelebA experiment.

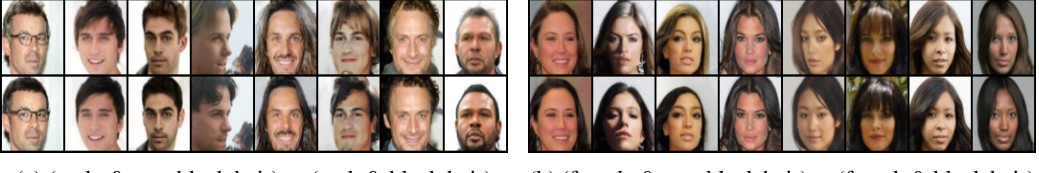

(a) (male & non-black hair) to (male& black hair)  (b) (female & non-black hair) to (female& black hair)

Figure 14: Majority to minority conversion through our objective in CelebA (Benchmark, 5%) experiment. The first row illustrates the samples from DSM(obs), and the second row illustrates the samples from TIW-DSM under the same random seed.

### E.4 DENSITY RATIO ANALYSIS

We provide more density ratio statistics according to diffusion time in various experiments which are discussed in Section 4.4. Figures 20 to 23 shows the case in FFHQ, and CelebA. Appendix E.4 shows the reweighting value on the 2-D cases.

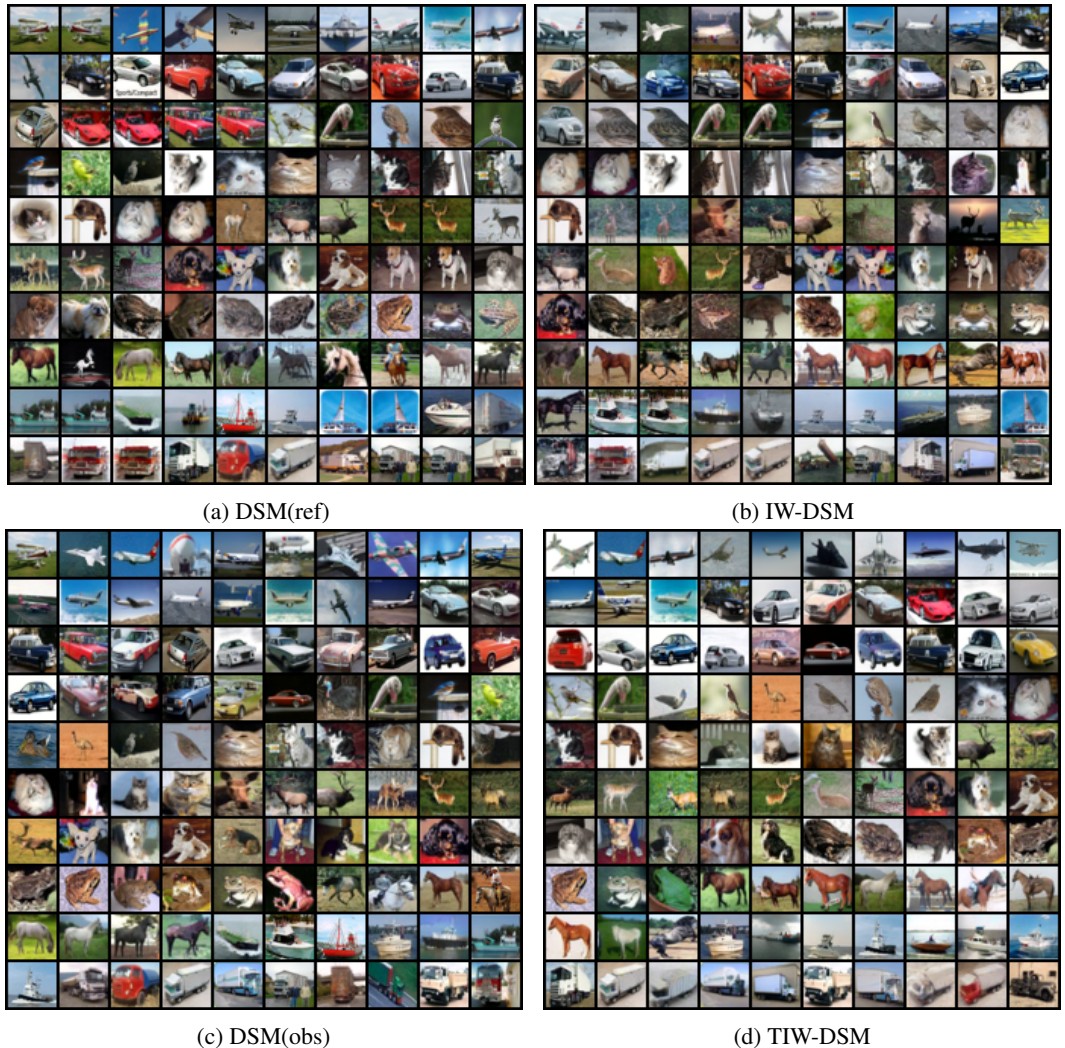

(a) DSM(ref)

(b) IW-DSM

(c) DSM(obs)

(d) TIW-DSM

Figure 15: Samples that reflect latent statistics from CIFAR-10 (LT / 5%) experiment.

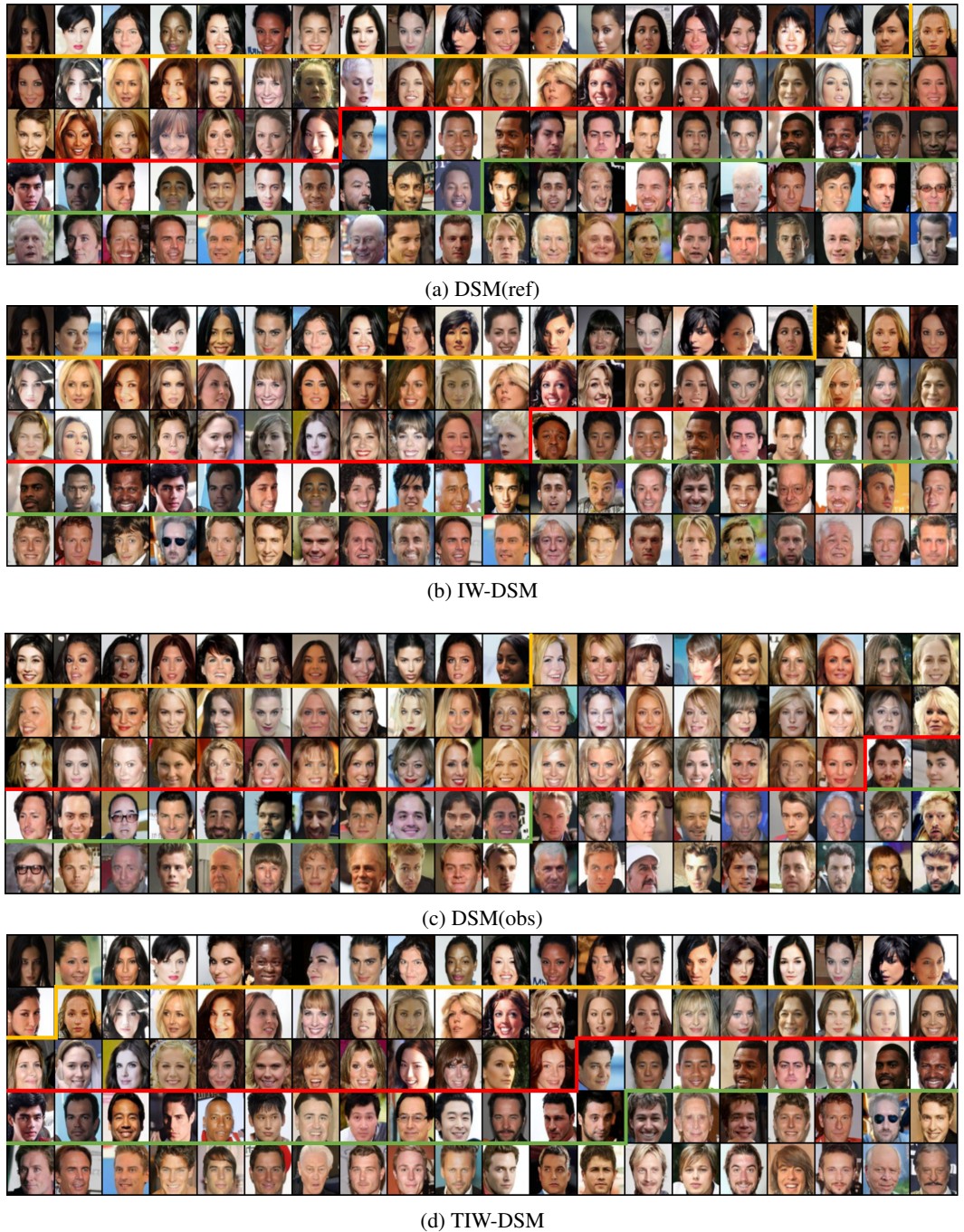

(a) DSM(ref)

(b) IW-DSM

(c) DSM(obs)

(d) TIW-DSM

Figure 16: Samples that reflect latent statistics from CelebA (Benchmark, 5%) experiment

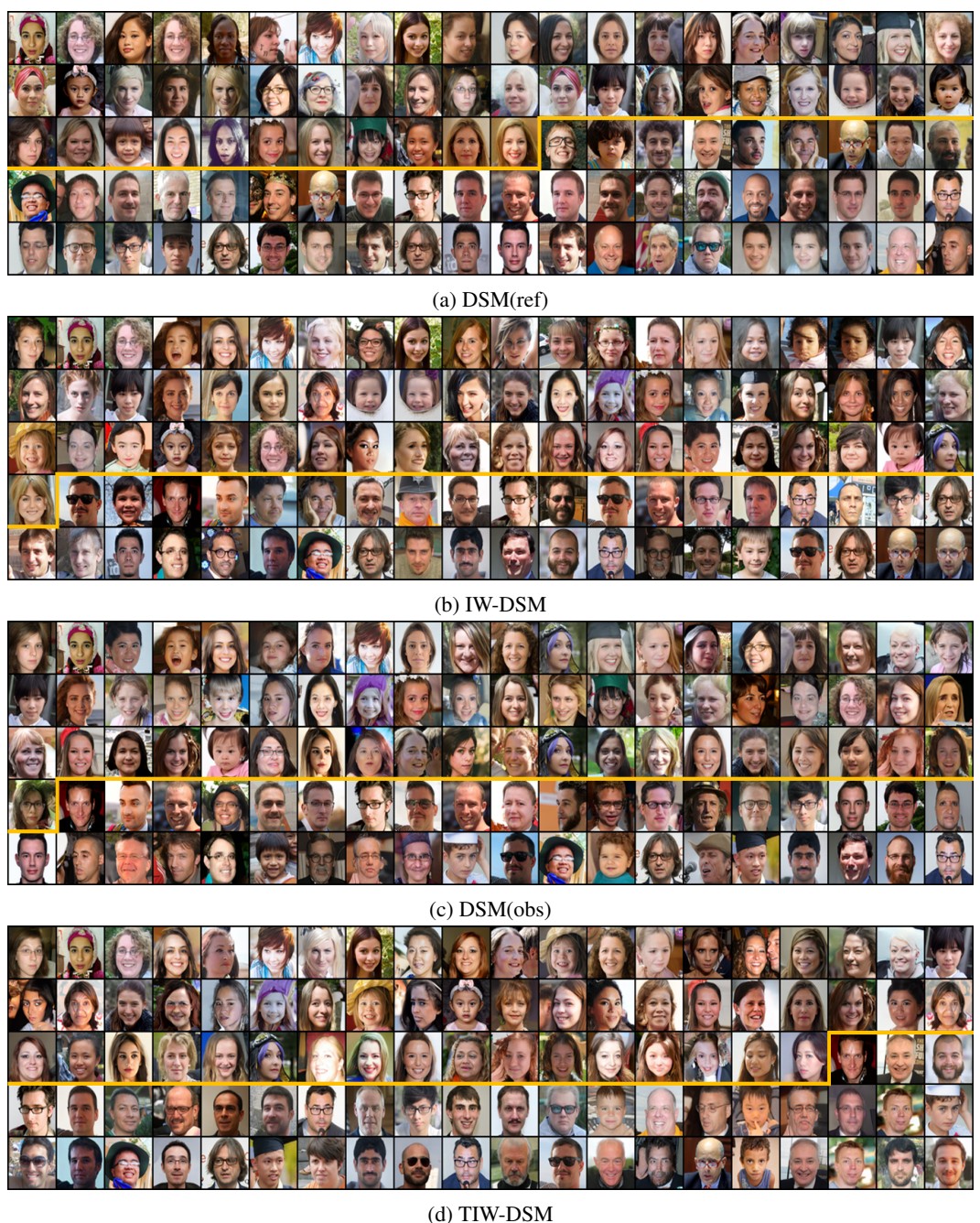

(a) DSM(ref)

(b) IW-DSM

(c) DSM(obs)

(d) TIW-DSM

Figure 17: Samples that reflect latent statistics from FFHQ (Gender 80%, 1.25%) experiment

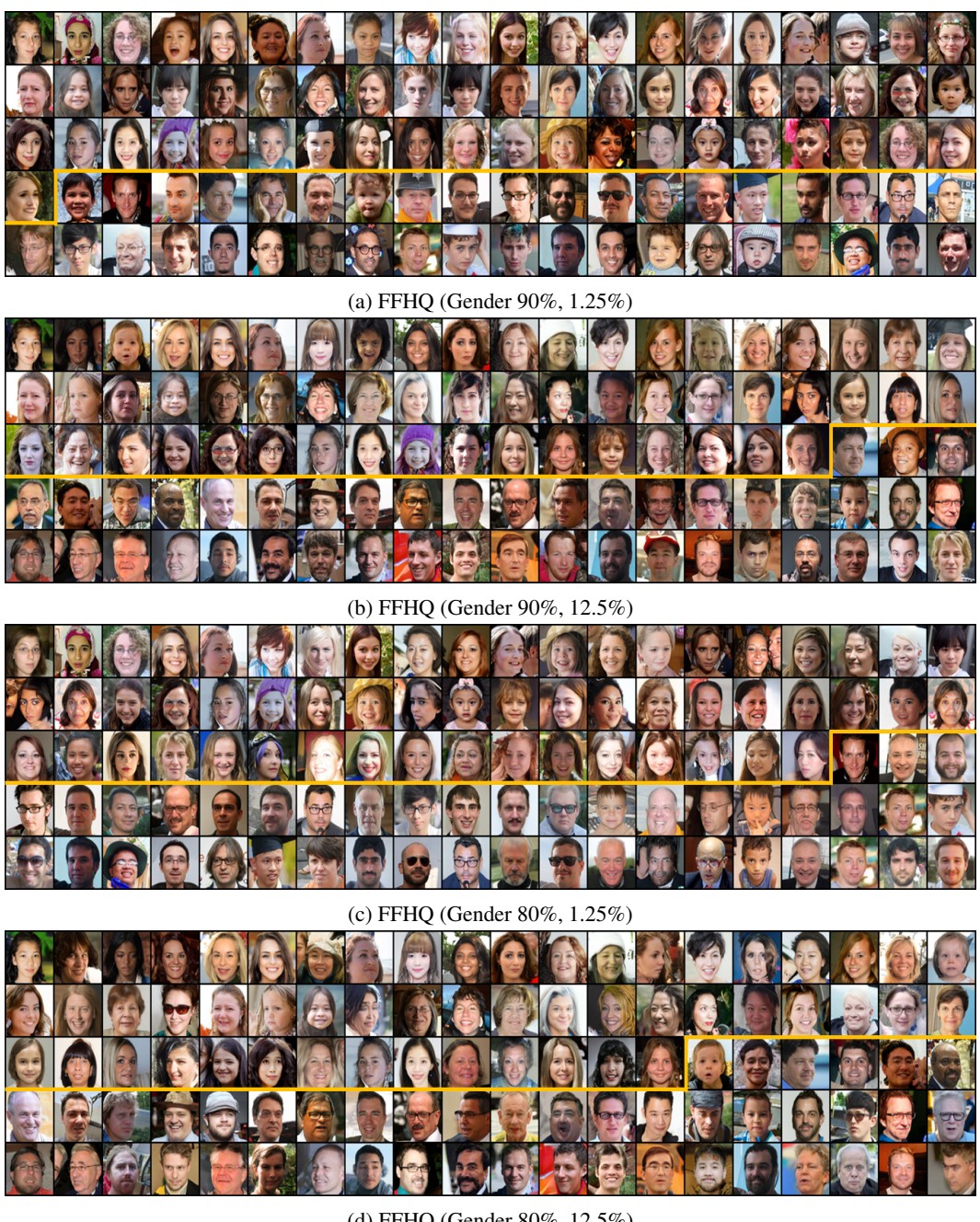

(a) FFHQ (Gender 90%, 1.25%)

(b) FFHQ (Gender 90%, 12.5%)

(c) FFHQ (Gender 80%, 1.25%)

(d) FFHQ (Gender 80%, 12.5%)

Figure 18: Samples that reflect latent statistics from TIW-DSM according to bias strength & reference size in FFHQ experiments.

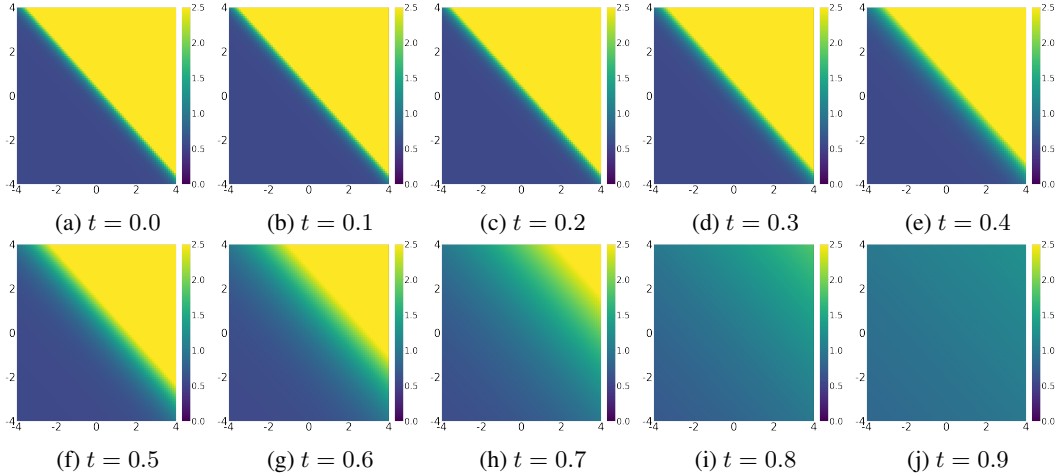

Figure 19: Density ratio analysis on 2-D example on various diffusion time (VP).

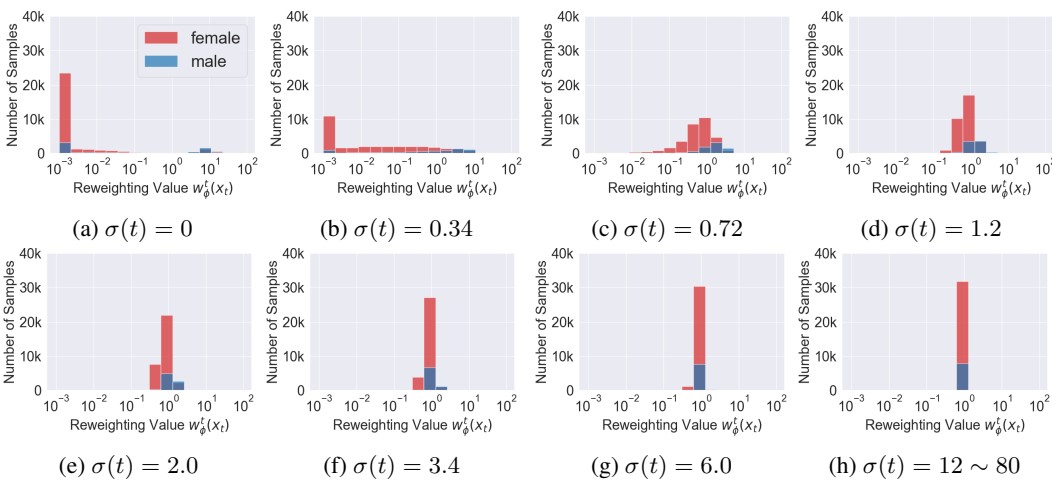

Figure 20: Density ratio analysis on FFHQ (Gender 80%, 12.5%) on various diffusion time for $\mathcal{D}_{\text{bias}}$

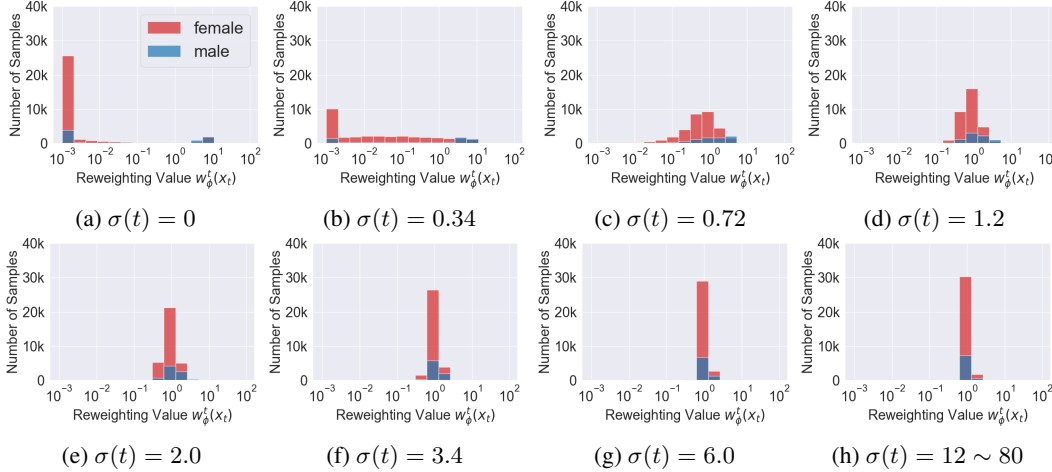

Figure 21: Density ratio analysis on FFHQ (Gender 90%, 12.5%) on various diffusion time for $\mathcal{D}_{\text{bias}}$.

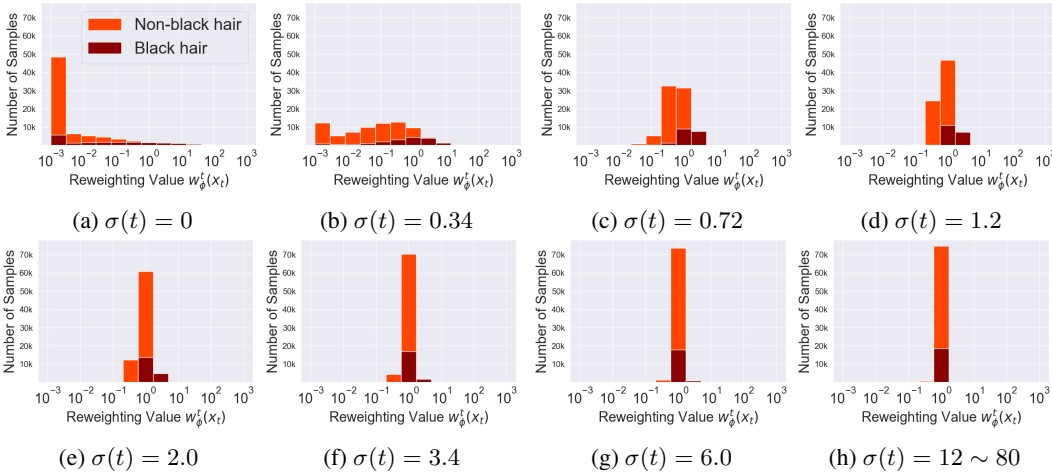

Figure 22: Density ratio analysis on CelebA (Benchmark, 5%) on various diffusion times. This figure only consider female in $\mathcal{D}_{\text{bias}}$.

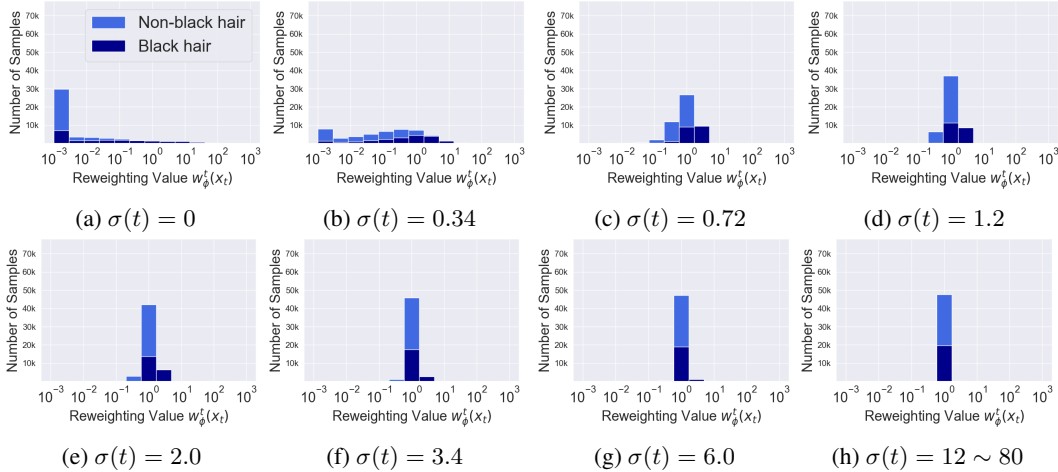

Figure 23: Density ratio analysis on CelebA (Benchmark, 5%) on various diffusion times. This figure only consider male in $\mathcal{D}_{\text{bias}}$.

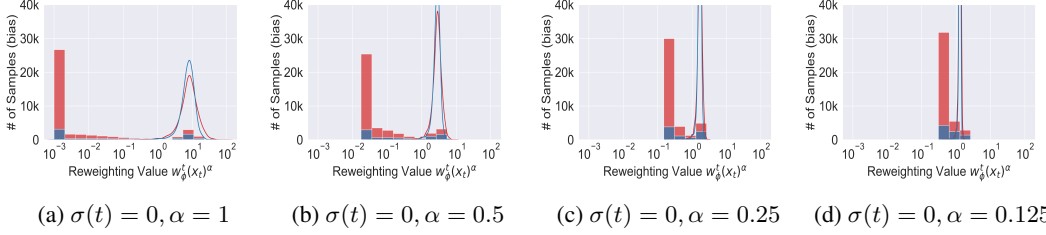

Figure 24: Density ratio analysis on FFHQ (Gender 80%, 12.5%) on zero diffusion time with ratio scaling for $\mathcal{D}_{\text{bias}}$ (box plot) and $\mathcal{D}_{\text{ref}}$ (density plot).

## E.5 EFFECTS OF DISCRIMINATOR ACCURACY

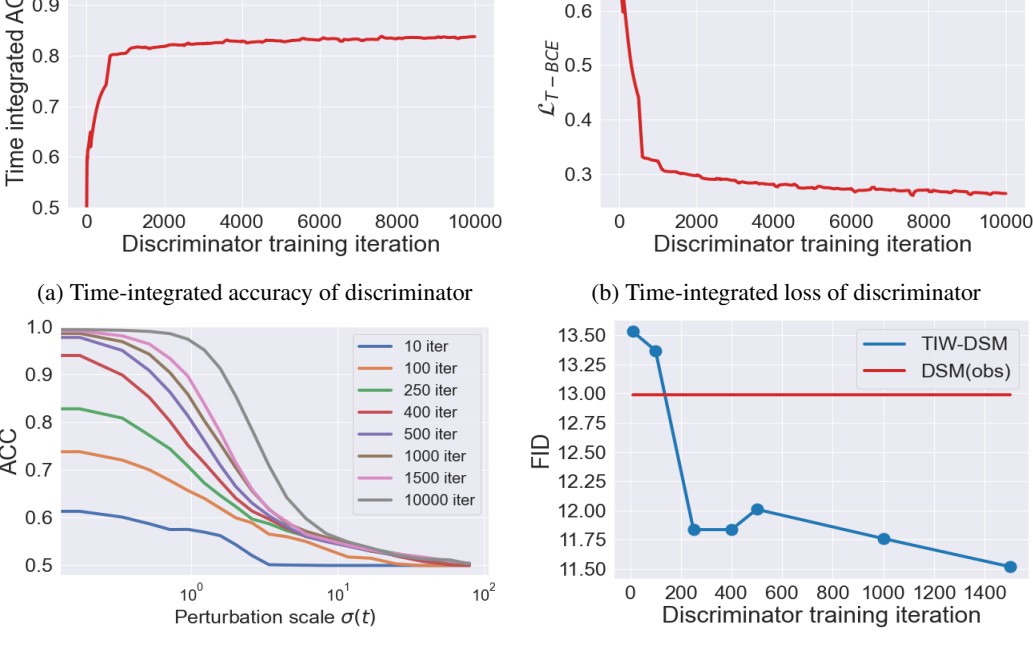

(a) Time-integrated accuracy of discriminator

(b) Time-integrated loss of discriminator

(c) Discriminator accuracy according to the time

(d) FID according to discriminator learning progress

Figure 25: Effects of discriminator accuracy on diffusion model training in CIFAR-10 (LT / 5%) experiments. The maturity of the time-dependent discriminator directly influences the performance of the diffusion model.

We analyze the learning progress of the time-dependent discriminator and its correlation with the diffusion model's performance. Figures 25a and 25b show the time-integrated accuracy and time-integrated loss value according to the discriminator training iteration. Note that perfect discrimination in terms of accuracy is impossible at a large perturbation scale. Figure 25c shows the accuracy according to $\sigma(t)$. As the training of time-dependent discriminator matures, accuracy improves for all perturbation scales.

Our objective, $\mathcal{L}_{\text{TIW-DSM}}(\boldsymbol{\theta}; p_{\text{bias}}, w_{\phi^*}^t(\cdot))$ assumes an optimal time-dependent discriminator, so analysis on maturity of the discriminator is important. Figure 25d shows the performance when training TIW-DSM using a less-trained discriminator. In the very early stages, the discriminator provides signals that are completely off, resulting in worse performance compared to DSM(obs). However, as it undergoes some level of training, it progressively enhances the performance of the diffusion model. The time-dependent discriminator with 1.5k iterations shows an FID of 11.52, which is nearly close to the reported value in the Table 1 of 11.51 obtained with 10k iterations. Note that the training 10k iterations of the discriminator only takes 30 minutes with 1 RTX 4090.

### E.6 COMPARISON TO THE GUIDANCE METHOD

A direct quantitative comparison with Friedrich et al. (2023) is infeasible because their approach is not based on a weak supervision setting. However, their method is based on the commonly used guidance method in the diffusion model, and it is possible to adapt the spirit of that method to our weak supervision scenario.

The unbiased data score $\nabla \log p_{\text{data}}(\mathbf{x}_t)$ can be represented by eq. (52), and it can be approximated with two neural networks as in eq. (53). $\alpha = 1$ for an ideal scenario, but it is usually adjusted for better performance. This is a similar mechanism in Section 4.3: Density ratio scaling. Table 9 and Figure 26 compare the guidance method and proposed method by adjusting $\alpha$. Note that the guidance method requires the evaluation of 2 neural networks for one denoising step, resulting in slow sampling.

$$\nabla \log p_{\text{data}}(\mathbf{x}_t) = \nabla \log p_{\text{bias}}(\mathbf{x}_t) + \nabla \log \frac{p_{\text{bias}}(\mathbf{x}_t)}{p_{\text{data}}(\mathbf{x}_t)} \tag{52}$$

$$\approx \mathbf{s}_{\boldsymbol{\theta}}(\mathbf{x}_t, t) + \alpha \nabla \log \frac{d_{\boldsymbol{\phi}}(\mathbf{x}_t, t)}{1 - d_{\boldsymbol{\phi}}(\mathbf{x}_t, t)} \tag{53}$$

Table 9: The comparison between the guidance method (Fair-Diffusion) and TIW-DSM for CIFAR-10 (LT / 5%) experiments.

|  | DSM(obs) | Fair-Diffusion | TIW-DSM |
|---|---|---|---|
| FID ($\alpha = 1$) | 12.99 | 12.55 | **11.51** |
| FID (optimal $\alpha$) | 12.99 | 12.15 | **11.51** |
| Sampling time | **7.5** Minute / 50k | 20.85 Minute / 50k | **7.5** Minute / 50k |

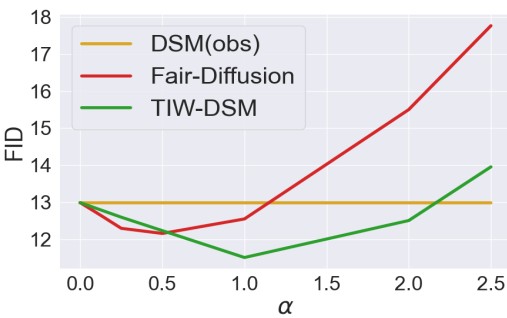

Figure 26: Comparison to guidance method (Fair-Diffusion) by adjusting $\alpha$ for CIFAR-10 (LT / 5%) experiments.

### E.7 OBJECTIVE FUNCTION INTERPOLATION

The time-dependent density ratio used in TIW-DSM is more precise than the time-independent density ratio, in the integration sense. However, the time-marginal density ratio from $w_{\phi^*}^t(\cdot)$ remains inaccurate for small diffusion time. One attractive direction is utilizing vanilla objective $\mathcal{L}_{\text{DSM}}(\boldsymbol{\theta}; p_{\text{bias}})$ for small diffusion time. Small diffusion time is known to be oriented towards denoising rather than addressing semantic information (Rombach et al., 2022; Xu et al., 2023), so objective interpolation is worth exploring.

We experimented with the preliminary approach, which interpolates the objectives as eq. (54). Note that $\sigma(\tau) = 0$ indicates the original TIW-DSM objective, and $\sigma(\tau) = 80$ indicates the vanilla DSM objective.

$$
\mathcal{L}_{\text{Interpolate}}(\boldsymbol{\theta}; p_{\text{bias}}, w_{\phi^*}^t(\cdot), \tau) \tag{54}
$$
$$
:= \frac{1}{2} \int_0^\tau \mathbb{E}_{p_{\text{bias}}(\mathbf{x}_0)} \mathbb{E}_{p(\mathbf{x}_t|\mathbf{x}_0)} \left[ \lambda(t) \left[ ||s_{\boldsymbol{\theta}}(\mathbf{x}_t, t) - \nabla \log p(\mathbf{x}_t|\mathbf{x}_0)||_2^2)\right] \right] \mathrm{d}t
$$
$$
+ \frac{1}{2} \int_\tau^T \mathbb{E}_{p_{\text{bias}}(\mathbf{x}_0)} \mathbb{E}_{p(\mathbf{x}_t|\mathbf{x}_0)} \left[ \lambda(t) w_{\phi^*}^t(\mathbf{x}_t) \left[ ||s_{\boldsymbol{\theta}}(\mathbf{x}_t, t) - \nabla \log p(\mathbf{x}_t|\mathbf{x}_0) - \nabla \log w_{\phi^*}^t(\mathbf{x}_t)||_2^2)\right] \right] \mathrm{d}t
$$

Contrary to intuition, the result in Figure 27 shows that it does not improve the performance but rather smoothly interpolates between two objectives. We suspect that a hard truncation between the objectives may not be the best choice. We consider the gradual change of the objective according to time as a future work.

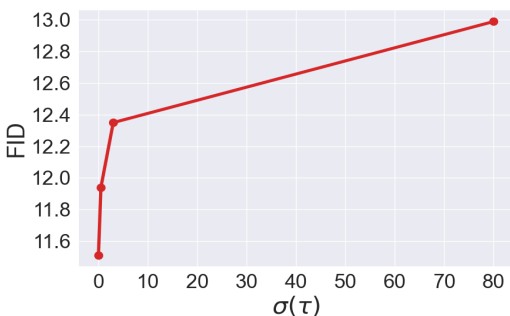

Figure 27: Objective interpolation according to $\sigma(\tau)$ in CIFAR-10 (LT / 5%) experiments.

## E.8 Fine tuning Stable Diffusion

The existing large-scale text-to-image diffusion model suffers from serious bias (Maggio, 2022). For example, if you type "nurse" in the prompt, only female nurses appear, as shown in Figure 28. Our method involves mitigating latent bias, so we can consider a scenario where we mitigate gender as a latent bias. We obtained a reference dataset of approximately 50 images and fine-tuned Stable Diffusion (Rombach et al., 2022) for the "nurse" prompt using the framework in Ruiz et al. (2023) with our objective TIW-DSM. The fine-tuned Stable Diffusion successfully generated a male nurse as shown in Figure 29.

We consider this to be the primary result of applying our objective to text-to-image diffusion models. In addition to fine-tuning, this approach can be applied to training a text-to-image model from scratch. Constructing a reference set for (prompt, bias) pairs deemed important by society and applying our objective during training, should enable a relatively fair generation.

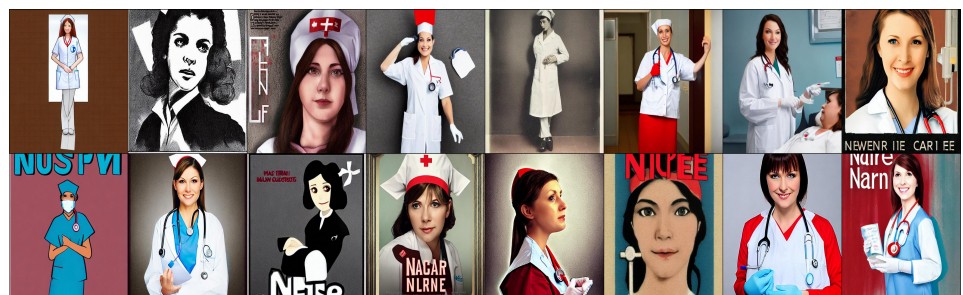

Figure 28: Samples from Stable Diffusion with prompt "nurse"

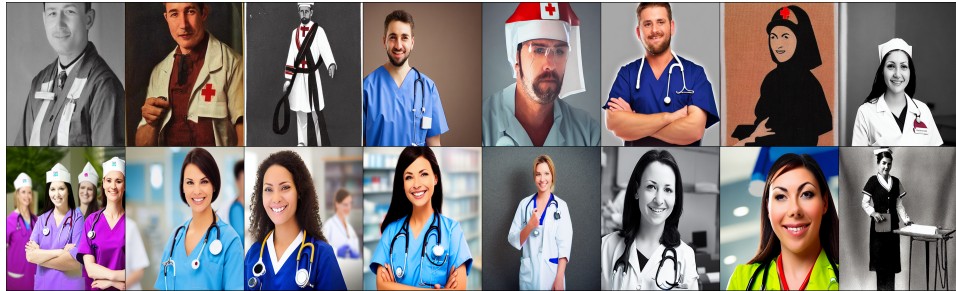

Figure 29: Samples from fine-tuned Stable Diffusion on prompt "nurse" with TIW-DSM.

### E.9 DATA AUGMENTATION WITH STABLE DIFFUSION

The baseline DSM(ref) does not exhibit good performance, because it suffers from a limited number of $\mathcal{D}_{\text{ref}}$, leading to poor diversity of generated samples. One consideration is to request Stable Diffusion to generate unbiased samples and use them in conjunction with $\mathcal{D}_{\text{ref}}$. We request Stable Diffusion to generate 500 samples with the prompt "a photo of man" and another 500 samples with the prompt "a photo of woman" and resizing them to fit our experiment setting in FFHQ as shown in Figures 30 and 31.

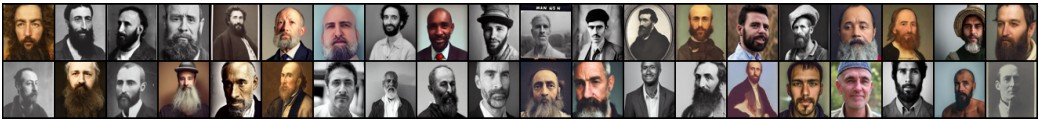

Figure 30: Samples from Stable Diffusion with prompt "a photo of man"

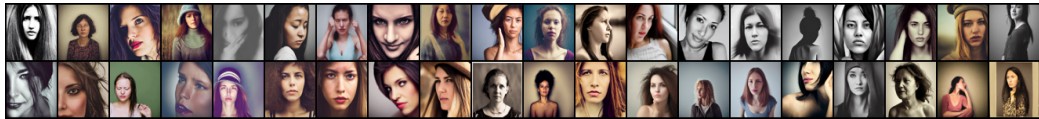

Figure 31: Samples from Stable Diffusion with prompt "a photo of woman"

Table 10 presents indirect results quantifying the data augmentation method using Stable Diffusion. The method SD represents the performance of directly generated samples from Stable Diffusion. It's noteworthy that SD exhibits poor performance at 95.63, primarily because unannotated biases such as age, and race are not controlled. On the other hand, DSM(ref) and TIW-DSM utilize statistics from balanced reference data, which is free from unannotated bias without point-wise supervision. This result underscores the reason why we should not rely solely on a large-scale foundation model. DSM(ref) + SD indicates the half of generated samples from DSM(ref) and the other half of the samples from SD. This can be considered as a performance similar to vanilla DSM training with $\mathcal{D}_{\text{ref}}$ and generated samples from SD. The performance of DSM(ref)+SD is poor due to a serious bias in Stable Diffusion samples.

Table 10: The effects of data augmentation with Stable Diffusion for FFHQ (80% / 12.5%) experiments.

| Method | FID 50k | FID 1k |
|---|---|---|
| DSM(ref) | 6.22 | 21.87 |
| SD | - | 95.63 |
| DSM(ref) + SD | - | 43.57 |
| TIW-DSM | **4.49** | **20.39** |

