# OpenReview forum: "Training Unbiased Diffusion Models From Biased Dataset"
_ICLR.cc/2024/Conference — ICLR 2024 poster_

### Official Review · Reviewer_irEP · 2023-10-28

**Soundness:** 3 good
**Presentation:** 3 good
**Contribution:** 3 good
**Rating:** 6
**Confidence:** 4

**Summary:**

This paper addresses the issue of dataset bias in diffusion models. Authors introduces a novel method called Time-dependent Importance reWeighting (TIW). This method, leveraging a time-dependent density ratio, offers improvements in both reweighting and score correction processes. One of its major contributions is making the weighting objective more manageable and drawing a theoretical link to traditional score-matching objectives that aim for unbiased distributions. Experiments shows TIW outperforms other baselines.

**Strengths:**

1. The problem this paper tries to address is important.

2. The methodology introduced is innovative.

3. This research creates a theoretical bridge between the proposed objective and the conventional score-matching objectives, offering a robust guarantee.

4. Extensive testing of the method across diverse datasets and under varied conditions underscores its robustness and versatility.

5. Preliminary results from the experiments are encouraging.

**Weaknesses:**

1. Introducing time-dependent methods might lead to heightened computational complexities in contrast to their time-independent counterparts. It would be beneficial to include a comparative analysis or discussion addressing this aspect.

2. One pivotal query that emerges is whether the efficiency and overall success of the proposed method are intrinsically tied to the performance of the time-dependent discriminator. It would be enlightening to elucidate this relationship.

3. There's a pertinent concern regarding the model's robustness. If the model isn't aptly regularized or subjected to limited training data, it might inadvertently heighten the risk of overfitting.

4. The research paper titled "Fair Diffusion" by Friedrich et al. [1] is notably aligned with the theme of fairness in diffusion models. However, its conspicuous absence in the current work, either as a benchmark or a referenced study, is intriguing. The authors might want to consider explicating their rationale behind not incorporating it as a baseline or offering a comparative analysis.


[1] Friedrich et al. "Fair Diffusion: Instructing Text-to-Image Generation Models on Fairness."

**Questions:**

Can this method be transferred to other models like GAN-based model?

---

> ### Author Response · Authors · 2023-11-17
> **Thank you for the questions and feedback**
>
> We appreciate the reviewer's sincere and helpful feedback. We answer for reviewer’s comments below.
>
> **Q1. [Computation Cost]** *Introducing time-dependent methods might lead to heightened computational complexities in contrast to their time-independent counterparts. It would be beneficial to include a comparative analysis or discussion addressing this aspect.*
>
> **Response to Q1.**
>
> Thank you for addressing important issues. We compare the computation cost of time-independent importance reweighting (IW-DSM), and time-dependent importance reweighting (TIW-DSM) in terms of 1) training time, 2) training memory, 3) sampling time, and 4) sampling memory.
>
> Both methods require the evaluation of the discriminator for diffusion model training. However, TIW demands 34% more training time and 13% more training memory compared to IW-DSM. This is because we need to compute a new term $\nabla \log{w^t_{\phi}}(x_t)$ using the automatic gradient module in the PyTorch, while IW-DSM only requires a feed-forward value. However, once the training is complete, the discriminator is not used for sampling, so the sampling time & memory remain the same. We conducted tests on RTX 4090 * 4 Cores on CIFAR-10 experiments. We also reflect it in Appendix D.5. Note that the training of time-dependent discriminator is negligibly cheap, converging around 10 minutes with 1 RTX 4090.
>
> ||IW-DSM|TIW-DSM|
> |---|---|---|
> |Training time|0.26 Second/Batch |0.34 Second/Batch|
> |Training memory|13,258 MiB*4Core|15,031 MiB*4Core|
> |Sampling time| 7.5 Minute/50k |7.5 Minute/50k|
> |Sampling memory| 4,928 MiB*4Core |4,928 MiB*4Core |

---

> > ### Author Response · Authors · 2023-11-17
> > **Continued**
> >
> > **Q2. [Effects of performance of time-dependent discriminator]** *One pivotal query that emerges is whether the efficiency and overall success of the proposed method are intrinsically tied to the performance of the time-dependent discriminator. It would be enlightening to elucidate this relationship.*
> >
> > **Response to Q2.**
> >
> > Thank you for your valuable feedback. We analyze the relations in terms of three viewpoints, A2.1) the influence of less-trained discriminator, A2.2) density ratio scaling, and A2.3) time independent vs time dependent discriminator.
> >
> > **A2.1 The influence of less-trained discriminator**: We analyzed the performance of the time-dependent discriminator and its effects on the diffusion model, and added Appendix E.5. Figure 25-(a,b) shows the training curve of the time-dependent discriminator, and Figure 25-(c) illustrates the time-wise accuracy. As the discriminator matures, the diffusion model training with TIW-DSM performs progressively better.  A discriminator trained for 1.5k iterations showed performance similar to a discriminator for 10k iterations for diffusion model training. Considering that the discriminator for 10k iterations takes around 30 minutes, the training of the discriminator is highly cost-effective.
> >
> > **A2.2 Density ratio scaling**: In Section 4.3: Density ratio scaling, Figure 7 illustrates when experimented using scaled ratio $w_{\phi*}^t(x_t)^{\alpha}=\big[\frac{p_{data}^t(x_t)}{p_{bias}^t(x_t)}\big]^{\alpha}$ instead of $w_{\phi*}^t(x_t)=\frac{p_{data}^t(x_t)}{p_{bias}^t(x_t)}$, which implies smoothing ($\alpha<1$) or sharpening ($\alpha>1$) the discriminator logit. In other words, we can consider the adjustment of $\alpha$ as a performance change in the discriminator. Theoretically, $\alpha \in [0,1] $ interpolates the original TIW-DSM and DSM(obs) baseline. This demonstrates a gradual evolution from DSM(obs) to TIW-DSM, showing simultaneous improvements in bias mitigation and FID. The sharpening with $\alpha > 1$ further mitigates bias to a certain extent, but FID incurs a slight trade-off. Note that the performance seems well-balanced under the theoretical optimal value of $\alpha=1$.
> >
> > **A2.3 Time-independent vs time-dependent discriminator**: As explained in Appendix A.3, the objective function using time-independent importance reweighting (IW-DSM) and time-dependent importance reweighting (TIW-DSM) is equivalent w.r.t. $\theta$, when the discriminator is optimal. Therefore, the sole source of performance gains is our accurate discriminator.

---

> > > ### Author Response · Authors · 2023-11-17
> > > **Continued**
> > >
> > > **Q3. [The robustness against overfitting]** *There's a pertinent concern regarding the model's robustness. If the model isn't aptly regularized or subjected to limited training data, it might inadvertently heighten the risk of overfitting.*
> > >
> > > **Response to Q3.**
> > >
> > > Thank you for your valuable feedback. Robust training against overfitting is necessary for practical uses. There are several sources of overfitting from an engineering perspective, such as network architecture, Exponential Moving Average (EMA), and diffusion noise scheduling [d1, d2, d3], as mentioned in Appendix C. These configurations vary across datasets, so there is no golden rule. While finding robust settings by adjusting each component is important, our paper primarily focuses on the utility of the improved objective function, making this a slightly divergent topic.
> > >
> > > From the perspective of the objective function, TIW-DSM outperforms the baselines (See Figure 4-(e), Figure 5, Figure 13-(a-d)) in terms of robustness. We have analyzed Figure 8 and Figure 10-(a) to understand the reasons for robustness, which TIW-DSM can effectively utilize the information from $D_{bias}$. Nevertheless, we acknowledge the issues you raised. We committed to allocating the remaining GPU resources to experiments aimed at achieving robust learning (by adjusting architecture, EMA, and noise scheduling) until the final draft.
> > >
> > > [d1] Song et al. “Improved techniques for training score-based generative models.” (Neurips 2020)
> > >
> > > [d2] Nichol et al. “Improved denoising diffusion probabilistic models.” (Neurips 2021)
> > >
> > > [d3] Noon et al. “Fine-tuning Diffusion Models with Limited Data” (Neurips workshop 2022)

---

> > > > ### Author Response · Authors · 2023-11-17
> > > > **Continued**
> > > >
> > > > **Q4. [Comparison to Fair diffusion]** *The research paper titled "Fair Diffusion" by Friedrich et al. [1] is notably aligned with the theme of fairness in diffusion models. However, its conspicuous absence in the current work, either as a benchmark or a referenced study, is intriguing. The authors might want to consider explicating their rationale behind not incorporating it as a baseline or offering a comparative analysis.*
> > > >
> > > > *[1] Friedrich et al. "Fair Diffusion: Instructing Text-to-Image Generation Models on Fairness."*
> > > >
> > > > **Response to Q4.**
> > > >
> > > > Thank you for your helpful feedback. While the paper “Fair Diffusion” you mentioned has not been published, we agree that comparing it with concurrent work enhances the solidity of our paper. We elucidate the distinctions between our paper and “Fair Diffusion” from the perspectives of the **problem setting** and **method**. Furthermore, we conduct a quantitative comparison with TIW-DSM and method from “Fair Diffusion” which adaptively apply to our problem setting. We also added a reference study in Appendix B.1. and quantitative comparison in Appendix E.6.
> > > >
> > > > **A4.1. problem setting**: We adopt a weak supervision setting, which only requires a few amounts of fair samples $D_{ref}$, and a point-wise latent bias factor is not required. On the other hand, “Fair Diffusion” is based on large-scale text-to-image pre-trained diffusion models, and mitigates the bias that relies on text prompts forwarding into diffusion models. This indicates they utilize explicit supervision in the form of (images, text) pairs. Note that our paper’s weak supervision is suitable for affordable dataset collection scenarios.
> > > >
> > > > **A4.2. method**: We categorize the score correction scheme in two ways in Appendix B.3. “Fair diffusion” falls under the category of guidance methods, while TIW-DSM belongs to score-matching regularization. This signifies a methodological departure, and we highlight that our method does not increase the sampling, unlike guidance methods.
> > > >
> > > > **A4.3. Compare with guidance method**: The direct quantitative comparison between “Fair Diffusion” and TIW-DSM is infeasible due to differences in problem settings. However, we can consider applying the spirit of their approach, which is the guidance method. We provide a detailed explanation of implementing "Fair Diffusion" and the results by adjusting the guidance scale in Appendix E.6. The following table is a brief summary of the experimental results.
> > > >
> > > > ||DSM(obs)|Fair Diffusion|TIW-DSM|
> > > > |---|---|---|---|
> > > > |FID ($\downarrow$) |12.99|12.55|11.51|
> > > > |Sampling time ($\downarrow$) |7.5 Minute/50k|20.85 Minute/50k|7.5 Minute/50k|

---

> > > > > ### Author Response · Authors · 2023-11-17
> > > > > **Continued**
> > > > >
> > > > > Q5. **[Connection to GAN backbones]** *Can this method be transferred to other models like GAN-based model?*
> > > > >
> > > > > **Response to Q5.**
> > > > >
> > > > > Thank you for your interesting question. Since the concept of diffusion time does not exist in GANs, making direct application is challenging. However, we can provide a promising direction for GANs.
> > > > >
> > > > > Current GAN utilizes diffusion processes for various purposes. [d4] employs the diffusion process to mitigate the mode collapse, and [d5] uses it for data augmentation. Note that these techniques are orthogonal to the previous developments on GANs, making them relatively freely available for any GAN implementation. [d4, d5] consider samples from perturbed data distributions, thus the proposed time-dependent importance reweighting naturally applicable builds upon those methods.
> > > > >
> > > > > The other direction involves utilizing improved estimation techniques for time-independent density ratio. A notably subpar density ratio technique has been investigated at the synthetic dataset level using the telescoping approach [d6], and its generalized version [d7]. It could serve as another approach to mitigate estimation errors in both time-independent density ratio as well as time-dependent density ratio.
> > > > >
> > > > > [d4] Xiao et al. “Tackling the generative learning trilemma with denoising diffusion gans (ICLR 2022)
> > > > >
> > > > > [d5] Wang et al. “Diffusion-gan: Training gans with diffusion” (ICLR 2023)
> > > > >
> > > > > [d6] Rhodes et al. “Telescoping density-ratio estimation” (Neurips 2020)
> > > > >
> > > > > [d7] Choi et al. “Density ratio estimation via infinitesimal classification” (Aistats 2022)

---

> ### Comment · Reviewer_irEP · 2023-11-22
> **Thank you**
>
> Thank you for your detailed rebuttals. After reading all the rebuttals, including those from other reviewers, I would like to adjust my initial score from 'borderline accept' (6) to 'weak accept' (7). Unfortunately, there isn't a specific selection for 'weak accept,' as the next available option is 'accept' with a score of 8. Therefore, I have decided to increase my confidence in my assessment to 4.
>
> To all ACs and SACs, please consider my suggestions as a 'weak accept.' Thank you for your understanding.

---

> > ### Author Response · Authors · 2023-11-22
> > **Thank you for your response!**
> >
> > Dear reviewer,
> >
> > We appreciate the positive correspondence you've shared and extend sincere thanks. Your insights hold significant value for us, and we will carefully consider them to improve the overall quality of our paper. Your dedication in sharing valuable perspectives is recognized, and if there are any uncertainties on your end, please feel free to inform us.

---

### Official Review · Reviewer_WcK7 · 2023-10-29

**Soundness:** 3 good
**Presentation:** 3 good
**Contribution:** 2 fair
**Rating:** 5
**Confidence:** 3

**Summary:**

This paper proposed a method to train an unbiased diffusion model with a weak supervision setting based on a sampling process that they called time-independent importance reweighting. They first analyzed why they should use the time-independent importance reweighting, then how to apply score matching to the reweighting process.

**Strengths:**

1. The paper builds on successful recent advances in diffusion models, which excel in high-fidelity image generation within generative learning frameworks. Thus, the proposed TIW method shows promise for diverse applications, including text-to-image generation, image-to-image translation, and video generation.
2. The paper addresses the understudied issue of dataset bias in generative modeling, introducing a method to enhance the fairness and reliability of ML systems.
3.The paper establishes a theoretical equivalence between the proposed method and existing score-matching objectives from unbiased distributions, which could provide a strong foundation for the validity of their approach.

**Weaknesses:**

1. The paper builds upon a large body of previous work, which while showing the paper’s relevance, also means that any limitations or issues in those previous works could potentially affect the validity and effectiveness of TIW. The actually implementation process is similar to previous work on GAN.
2. The paper shows that it mitigates the bias in the dataset, but I am not sure whether that method would influence the overall generation quality. It is important to see detailed qualitative and quantitative analysis.

**Questions:**

N/A

---

> ### Author Response · Authors · 2023-11-17
> **Thank you for the questions and feedback**
>
> We appreciate the reviewer's feedback. We answer for reviewer's comments below.
>
> **[Clarification on miscommunicated point]**
>
> In response to the reviewer's summary, we would like to kindly point out that there are some miscommunicated points. The proposed method, TIW-DSM in eq. (10) is based on time-dependent importance reweighting, which is distinctly different from IW-DSM which leverages time-independent importance reweighting in eq. (7). The reviewer seems to think of our method as IW-DSM, but IW-DSM is the baseline method. We will consider distinguishing the two methods with more easily understandable names for the final draft.

---

> > ### Author Response · Authors · 2023-11-17
> > **Continued**
> >
> > **Q1. [Comparison to the method from GAN]** *The paper builds upon a large body of previous work, which while showing the paper’s relevance, also means that any limitations or issues in those previous works could potentially affect the validity and effectiveness of TIW. The actually implementation process is similar to previous work on GAN.*
> >
> > **Response to Q1.**
> >
> > Thank you for your feedback. We emphasize that the method proposed based on GAN [c1] is time-independent importance reweighting, and direct application [c1] to the diffusion model is described in Eq. (7), also known as IW-DSM in this paper.
> >
> > We highlight the limitation of time-independent importance reweighting in Section 3.1, which suffers from the density chasm problem. Figure 2-(e) illustrates that the estimation of the density ratio at $t=0$ is erroneous and becomes accurate as $t$ increases. Motivation by this, the proposed method employs time-dependent importance reweighting to alleviate the error propagation observed in the previous method. It is noteworthy that the importance reweighting through the time-dependent scheme is our own distinctive approach, as mentioned by reviewer 6Mt9 as well.
> >
> > The actual implementation process of TIW-DSM is different from that of IW-DSM, primarily because of the time-dependent reweighting objective in eq. (9) is intractable. To address this issue, we introduce eq. (10) as a tractable form with Theorem 1. In eq. (10), the time-dependent density ratio simultaneously addresses bias as a weighting mechanism and score correction. This is the reason for the difference in the implementation form compared to previous work.
> >
> > [c1] Choi et al. Fair generative modeling via weak supervision (ICML 2020).

---

> > > ### Author Response · Authors · 2023-11-17
> > > **Continued**
> > >
> > > **Q2. [The quantitative & qualitative analysis]** *The paper shows that it mitigates the bias in the dataset, but I am not sure whether that method would influence the overall generation quality. It is important to see detailed qualitative and quantitative analysis.*
> > >
> > > **Response to Q2.**
> > >
> > > Thank you for your feedback. We test our method across diverse datasets (CIFAR-10 / CIFAR-100 / FFHQ / CelebA) under various weak supervision settings, as mentioned by reviewer irEP as well. As described in Section 4, the major metric we utilize is FID between generated samples and unbiased datasets. Note that FID is the most representative metric to quantify sample quality and sample statistics. Indeed, superior FID indicates both good sample quality and good bias mitigation performance. Table 1~3 quantitatively compares the proposed method and baselines in terms of FID. Figure 7 illustrates the Bias (bias mitigation quality) and FID (sample quality + bias mitigation quality) tradeoff plots, and the proposed method is superior in bias mitigation as well as sample quality. For the qualitative sample analysis, please refer to Figure 15-18 to see the generated samples. Note that these samples reflect latent bias statistics as well as sample quality.

---

> > > > ### Author Response · Authors · 2023-11-23
> > > > **Do our revisions and responses answer your concerns and questions?**
> > > >
> > > > Dear Reviewer WcK7,
> > > >
> > > > We thank the Reviewer for the constructive comments. As the end of the discussion period is approaching (12 hours left), we would like to ask whether our paper revisions and responses have addressed your concerns and questions adequately. If not, we would be happy to discuss and update our paper further.
> > > >
> > > > Regards,
> > > >
> > > > the authors.

---

### Official Review · Reviewer_6Mt9 · 2023-11-01

**Soundness:** 3 good
**Presentation:** 4 excellent
**Contribution:** 3 good
**Rating:** 8
**Confidence:** 4

**Summary:**

The paper proposes time-dependent importance reweighting to mitigate the dataset bias for diffusion training. The proposed method utilizes time-dependent density ratio estimation, which is more precise than the previous time-independent one used in GAN. The authors provide both theoretical understanding and empirical demonstrations of the proposed method's benefits.

**Strengths:**

1. The paper is very well-written and easy to follow. I really enjoy the presentation flow and the thought process from Eq.7 to Eq.9 to Eq.10, with clear reasoning behind each change.
2. The method is intuitive and simple, yet clever and effective. Utilization of importance sampling by time-dependent density ratio estimation (DRE) in diffusion models is novel and natural. DRE becomes more precise as $t$ becomes larger.
3. I also appreciate the authors' efforts in providing toy examples and figures, which further makes the paper easy to understand.
4. The problem the paper proposes to study is an important one, as real-world datasets are never short of biases, especially the unannotated ones.
5. The proposed method's empirical performance is convincing. I also appreciate the inclusion of Eq.7 training numbers for the ablation study.

**Weaknesses:**

I do not have any major complaints. I list my minor questions and suggestions in the Questions section below.

**Questions:**

1. I can see the arguments for time-dependent density ratio estimation the same as the ones for [1] (especially), [2], etc. Thus I suggest the authors make the citations and discussions accordingly. Nonetheless, I agree that the use case of DRE is different.
2. In theory, the method works if $E_{x_0 \sim p_{\textrm{bias}}(x_0|x_t)} [\nabla\log p(x_t|x_0)] = \nabla\log p_{\textrm{bias}}^t(x_t)$ and DRE's gradient actually contains $\nabla\log p_{\textrm{data}}^t(x_t)$ information. Both require DRE to be really accurate, which, as the authors demonstrate, is reasonably true only for large $t$. Thus, might I suggest that the authors only train with Eq.10 for large $t$, and revert back (gradually) to standard diffusion training on the biased dataset for small $t$? The intuition is that for small $t$, the model only does denoising, which does not care about dataset biases (the model has already decided on the content it wants to generate). For an intuitive understanding, see Fig.2 in [3]. For a more theoretical explanation see [4], where small $t$'s score field is dominated by one sample, making the score for unbiased the same as biased here.
3. In Fig.4, every method's performance peaks early and gets worse. This is unexpected for me. Could the authors provide an explanation?


[1] Wang et al. "Diffusion-gan: Training gans with diffusion." ICLR 2023.

[2] Arjovsky et al. "Towards principled methods for training generative adversarial networks." ICLR 2017.

[3] Rombach et al. "High-resolution image synthesis with latent diffusion models." CVPR 2022.

[4] Xu et al. "Stable target field for reduced variance score estimation in diffusion models." ICLR 2023.

---

> ### Author Response · Authors · 2023-11-17
> **Thank you for the questions and feedback**
>
> We appreciate the reviewer for a deep understanding of our work and highly constructive suggestions. We answer the reviewer’s comments below.
>
> **Q1. [Time-dependent density ratio in GANs]** *I can see the arguments for time-dependent density ratio estimation the same as the ones for [1] (especially), [2], etc. Thus, I suggest the authors make the citations and discussions accordingly. Nonetheless, I agree that the use case of DRE is different.*
>
> *[1] Wang et al. "Diffusion-gan: Training gans with diffusion." ICLR 2023.*
>
> *[2] Arjovsky et al. "Towards principled methods for training generative adversarial networks." ICLR 2017.*
>
> **Response to Q1.**
>
> We appreciate your valuable advice. We acknowledge the existence of several works that employ noise injection tricks in the discriminator input of GANs. We have thoroughly examined the papers you provided guidance on, along with additional relevant papers, and have summarized our findings in Appendix B.4.

---

> ### Author Response · Authors · 2023-11-17
> **Continued**
>
> **Q2. [Objective function interpolation]** *In theory, the method works if $E_{x_{0}\sim p_{bias}(x_{0}|x_{t})} [\nabla \log p(x_{t}|x_{0})] = \nabla \log p_{bias}^{t}(x_t)$ and DRE's gradient actually contains $\nabla \log p_{data}^{t}(x_t)$ information. Both require DRE to be really accurate, which, as the authors demonstrate, is reasonably true only for large $t$. Thus, might I suggest that the authors only train with Eq.10 for large $t$, and revert back (gradually) to standard diffusion training on the biased dataset for small $t$? The intuition is that for small $t$, the model only does denoising, which does not care about dataset biases (the model has already decided on the content it wants to generate). For an intuitive understanding, see Fig.2 in [3]. For a more theoretical explanation see [4], where small $t$'s score field is dominated by one sample, making the score for unbiased the same as biased here.*
>
> *[3] Rombach et al. "High-resolution image synthesis with latent diffusion models." CVPR 2022.*
>
> *[4] Xu et al. "Stable target field for reduced variance score estimation in diffusion models." ICLR 2023.*
>
> **Response to Q2.**
>
> We appreciate your sincere suggestions. We acknowledge your suggestion makes sense and have conducted experiments in Appendix E.7. We attempted hard truncation of two objectives at a specific time $\tau$ as described in eq. (54), which utilize $(0,\tau)$ for $L_{DSM}$ objective, and $(\tau, T)$ for $L_{TIW-DSM}$ objective. Contrary to the intuition, the performance in Figure 27 is just interpolated. One possible reason is the negative effects of hard truncation on the objectives. Giving a sharp difference at the boundary of $\tau$ might not have been favorable for the generation path. As you suggested, we are considering experiments involving the gradual interpolation of the two objectives and will include the results in the final draft.

---

> ### Author Response · Authors · 2023-11-17
> **Continued**
>
> **Q3. [Overfitting on CIFAR-10 experiments]** *In Fig.4, every method's performance peaks early and gets worse. This is unexpected for me. Could the authors provide an explanation?*
>
> **Response to Q3.**
>
> As we described in Appendix C, some literature reports that overfitting is correlated with training configurations in network architecture, EMA, and diffusion noise scheduling. One of our own observations is that overfitting becomes more pronounced as the number of data decreases, as shown in Figure 10.
>
> Figure 4-(e) indicates that TIW-DSM shows less overfitting compared to IW-DSM. IW-DSM can not utilize the information from $D_{bias}$ effectively because time-independent density ratio provides extremely small reweighting value as shown in Figure 8-(a). On the other hand, TIW-DSM utilizes the information from $D_{bias}$ more effectively as shown in Figure 8-(b-d).

---

> ### Comment · Reviewer_6Mt9 · 2023-11-22
>
> I want to thank the authors for addressing my questions, especially in Appendix E.7. The results are indeed surprising to me, and perhaps they suggest that the quality of DRE is actually good when heavily sharing parameters despite the difficulty of the task, or the existing understanding of the score field is incomplete.
>
> I have read other reviewers' comments and the corresponding rebuttal, and I thank the authors for providing detailed responses. I still lean towards acceptance and thus keep my rating.

---

> > ### Author Response · Authors · 2023-11-22
> > **Thank you for your response!**
> >
> > Dear reviewer,
> >
> > Thank you for the valuable comments on the manuscript. Please let us know if you have further inquiries about the paper.
> >
> > Sincerely,
> >
> > Authors.

---

### Official Review · Reviewer_EYEW · 2023-11-03

**Soundness:** 2 fair
**Presentation:** 3 good
**Contribution:** 2 fair
**Rating:** 5
**Confidence:** 4

**Summary:**

The paper proposes a time-dependent importance-sampling objective for diffusion models, to mitigate the bias in the training set. The idea is to estimate the time-dependent importance weights between the bias distribution and the true (unbiased) distribution. The importance-weighted score-matching objective is then translated into a corresponding denoising score-matching objective. Controlled experiments showcase the effectiveness of the proposed method.

**Strengths:**

- The paper proposes a new denoising score-matching objective to tackle the biases in the training set.

- In contrast to the previous method (Eq.7), the proposed method estimates the time-dependent density ratio, which is arguably easier to estimate.

- Experiments on CIFAR-10/CIFAR-100/FFHQ demonstrate that the proposed method (TIW-DSM) not only outperforms baselines with solely reference unbiased data but also beats the vanilla go-to method IW-DSM.

**Weaknesses:**

- The biggest concern I have is that the proposed methods lack practical significance, and seem tangent to practical settings. The biases problem studies in this paper go away when considering conditional generation (e.g., text-to-image generation). For example, we can generate a batch of male images by conditioning on the "male" label, even though the female images are the majority.


- Regarding comparison to baseline (IW-DSM), some of the claimed advantages are not rigorous：(1) On page 4, "the perturbation provides an infinite number of samples from each distribution.": note that the perturbed distribution is $p_{data} * N(0, \sigma_t^2)$, one would still need large amount of data in $p_{data}$ (or $D_{ref}$) in order to estimate quantities related to $p_{data} * N(0, \sigma_t^2)$. (2) It's true that the accumulated MSE error in the density ratio estimation is smaller in the proposed method, but it introduces an additional erroneous term $\nabla \log w$ in the new objective. It could result in an even larger error compared to the baseline.

- Fig. 8.(a): I think a better density ratio estimator can get away from this over-confidence issue. A simple method is to regularize the classifier such that it cannot perfectly distinguish the two distributions.


- A natural baseline missing in the paper is asking generative models (e.g. SDXL) to generate more unbiased data for the considered small datasets and train the model in the vanilla DSM way.

## Update

I would like to thank the authors for the new experiments. The experiments showcase some promising results on applying the debiasing techniques to Stable Diffusion. However, one could argue that the chosen mode is too large (i.e., nurse). If the input prompt is "male nurse", then the problem still goes away without any fine-tuning. In addition, I could not immediately see why introducing the erroneous term $\nabla \log w$ would not negatively affect the training.

I raised my score to 5 based on the new experiments.

**Questions:**

- For the experiments, do the authors generate 50k images to calculate the FID score? What's the architecture and noisy schedule used in this paper? The current FID score seems quite worse off.

- How can this method be used in practical text-to-image generation?

---

> ### Author Response · Authors · 2023-11-17
> **Thank you for the questions and feedback**
>
> We appreciate the reviewer for the constructive and thoughtful feedback. We answer for reviewer’s comments below.
>
> **Q1. [Connection to text-to-image model]** *The biggest concern I have is that the proposed methods lack practical significance, and seem tangent to practical settings. The biases problem studies in this paper go away when considering conditional generation (e.g., text-to-image generation). For example, we can generate a batch of male images by conditioning on the "male" label, even though the female images are the majority*.
>
>
> **Response to Q1.**
>
> **[Mitigate bias in text-to-image models]** Thank you for your feedback. From the perspective of providing a text-to-image foundation model for practical use, addressing bias becomes a more crucial consideration. For instance, providing “nurse” as a condition results in the generation of only female nurses as shown in [a1] and Figure 28 in our paper. The proposed method introduces an approach to mitigate latent bias, so the proposed method could be applicable in this case. We constructed a reference dataset and fine-tuned the Stable Diffusion on the “nurse” prompt to mitigate bias on gender. The fine-tuned Stable Diffusion with the TIW-DSM objective successfully generates a certain proportion of male nurses as shown in Figure 29 (Please refer to Appendix E.8 for detailed explanations). This experiment demonstrates that our method works well in text-to-image diffusion models. In addition to fine-tuning, this approach can be applied to training a text-to-image model from scratch. Constructing a reference set for (prompt, bias) pairs deemed important by society and applying our objective during training, should enable a relatively fair generation.
>
> **[Dataset setting]** Indeed, text-to-image models are trained based on explicit supervision of (text, image) pairs. On the other hand, the weak supervision setting we employ is cost-effective in terms of dataset collection. While we acknowledge the role of general-purpose models like Stable Diffusion, it is also crucial to develop task-specific models starting from the dataset collection phase. It’s worth noting that dataset collection with weak supervision is prevalent in companies [a2].
>
> [a1] Maggio et al. The Bias problem: Stable Diffusion from https://vittoriomaggio.medium.com/the-bias-problem-stable-diffusion-607aebe63a37
>
> [a2] 23 & me. The Real Issue: Diversity in Genetics Research retrieved from https://blog.23andme.com/articles/the-real-issue-diversity-in-genetics-research.

---

> > ### Author Response · Authors · 2023-11-17
> > **Continued**
> >
> > **Q2. [Infinite number of samples]** *On page 4, "the perturbation provides an infinite number of samples from each distribution.": note that the perturbed distribution is $p_{data}$ * $N(0,σ_{t}^2)$, one would still need large amount of data in $p_{data}$ (or $D_{ref}$) in order to estimate quantities related to $p_{data}$ * $N(0,σ_{t}^2)$.*
> >
> > **Response to Q2.**
> >
> > Thank you for your valuable feedback. We acknowledge the presence of finite sample bias in $p_{data}$. However, rather than relying solely on the use of $p_{data}$ (in the case of time-independent density ratio estimation), perturbation can effectively reduce the Monte Carlo error. We refined our expression to focus on the reduction of Monte Carlo error (Please refer to Section 3.1 for the revised expression).

---

> > > ### Author Response · Authors · 2023-11-17
> > > **Continued**
> > >
> > > **Q3. [Accumulated MSE]** *It's true that the accumulated MSE error in the density ratio estimation is smaller in the proposed method, but it introduces an additional erroneous term ∇ log⁡ w in the new objective. It could result in an even larger error compared to the baseline.*
> > >
> > > **Response to Q3.**
> > >
> > > Thank you for your feedback. In fact, both TIW-DSM and IW-DSM are equivalent to $L_{SM}(\theta;p_{data})$ w.r.t. $\theta$ when considering optimal discriminator (See Appendix A.3). Thus, the difference between the value of $L_{SM}(\theta;p_{data})$ and each approximated objective value would be a straightforward comparison. However, since each is approximated in a form that includes a constant independent of $\theta$, direct comparison of objective values becomes intractable.
> > >
> > > The maturity of the discriminator in TIW-DSM and IW-DSM is the unique factor that leads to a better approximation of $L_{SM}(\theta; p_{data})$. Therefore, the consistent performance gains observed in CIFAR-10 / CIFAR-100 / FFHQ imply that the time-dependent discriminator approximating objectives align more closely with $L_{SM}(\theta; p_{data})$.

---

> > > > ### Author Response · Authors · 2023-11-17
> > > > **Continued**
> > > >
> > > > **Q4. [Regularizing time-independent density ratio]** *Fig. 8.(a): I think a better density ratio estimator can get away from this over-confidence issue. A simple method is to regularize the classifier such that it cannot perfectly distinguish the two distributions.*
> > > >
> > > > **Response to Q4.**
> > > >
> > > > Thank you for your insightful comments. In fact, we conducted an ablation study in Section 4.3: Density ratio scaling to adjust the density ratio for both IW-DSM and TIW-DSM, making it smoother or sharper. In other words, this ablation uses $w_{\phi*}^t(x_t)^{\alpha}=\big[\frac{p_{data}^t(x_t)}{p_{bias}^t(x_t)}\big]^{\alpha}$ rather than $w_{\phi*}^t(x_t)=\frac{p_{data}^t(x_t)}{p_{bias}^t(x_t)}$. This scaling was applied to the time-independent density ratio in the same way. The scaling strength $\alpha<1$ makes the density ratio becomes smoother as you suggested. The red line in Figure 7-(a, b) can be considered as a regularized density ratio for a time-independent discriminator. The performance of FID and bias mitigation ability is linearly changing with the strength of regularization $\alpha$. You can refer to Figure 24, which is a scaled version of Figure 8-(a), for the analysis of the scaled time-independent density ratio. Note that this scaling technique is orthogonally adaptable to TIW-DSM as well, and TIW-DSM shows better trade-offs in the change of $\alpha$.
> > > >
> > > > The exploration of a more effective density ratio estimator constitutes its own research topic. [a3] proposes a method of interpolating two distributions at appropriate intervals to estimate the ratio with multiple discriminators, and [a4] expands such bridges to an infinite number of intervals. Although these estimators require more inference time, they can be applied to both IW-DSM and TIW-DSM for better ratio estimation, respectively. We consider this as a future work. One clear point is that a better density ratio estimator can orthogonally be applied to the time-dependent density ratio, so the value of time-dependent importance reweighting remains intact.
> > > >
> > > > [a3] Rhodes et al. Telescoping density-ratio estimation (Neurips 2020)
> > > >
> > > > [a4] Choi et al. Density ratio estimation via infinitesimal classification (Aistats 2022)

---

> > > > > ### Author Response · Authors · 2023-11-17
> > > > > **Continued**
> > > > >
> > > > > **Q5. [Data augmentation with Stable Diffusion]** *A natural baseline missing in the paper is asking generative models (e.g. SDXL) to generate more unbiased data for the considered small datasets and train the model in the vanilla DSM way.*
> > > > >
> > > > > **Response to Q5.**
> > > > >
> > > > > Thank you for your thoughtful comments. We think the baseline you proposed addresses the issue of the limited data problem that was a drawback of vanilla DSM training using $D_{ref}$. However, it is important to note that the text-to-image generative model inherently contains bias. We experiment on data augmentation with Stable Diffusion based on FFHQ experiments (Please refer to Appendix E.9. for details about experiments). Although Stable Diffusion generates images equally for both genders, the performance significantly degraded due to the differences in age, race, and other factors compared to the reference set statistics. This result underscores the reason why we should not rely solely on a large-scale foundation model.
> > > > >
> > > > >
> > > > > |Method|FID 50k|FID 1k|
> > > > > |---|---|---|
> > > > > |DSM(ref)|6.22 |21.87|
> > > > > |SD|-|95.63|
> > > > > | DSM(ref) + SD| - |43.57|
> > > > > |TIW-DSM| **4.49** |**20.39** |

---

> > > > > > ### Author Response · Authors · 2023-11-17
> > > > > > **Continued**
> > > > > >
> > > > > > **Q6. [Experiment setting]** *For the experiments, do the authors generate 50k images to calculate the FID score? What's the architecture and noisy schedule used in this paper? The current FID score seems quite worse off.*
> > > > > >
> > > > > > **Response to Q6.**
> > > > > >
> > > > > > Thank you for your question. As described in Section 4: Metric, we use 50,000 generated samples to compute FID. Appendix D.2. provides detailed training configurations, and we adopt the best setting from EDM [a5]. For example, in CIFAR-10, the reason why the FID appears to be quite worse off is that we use a subset of the entire CIFAR-10 to construct the bias set $D_{bias}$. Please refer to Table 5 for our dataset construction. Note that the FID gets worse if we utilize a smaller dataset as shown in Figure 10-(b). This is why we present DSM(ref) and DSM(obs) as baselines.
> > > > > >
> > > > > > [a5] Karras et al. Elucidating the design space of diffusion-based generative models. (Neurips 2022)

---

> > > > > > > ### Author Response · Authors · 2023-11-23
> > > > > > > **Do our revisions and responses answer your concerns and questions?**
> > > > > > >
> > > > > > > Dear Reviewer EYEW,
> > > > > > >
> > > > > > > We thank the Reviewer for the constructive comments. As the end of the discussion period is approaching (12 hours left), we would like to ask whether our paper revisions and responses have addressed your concerns and questions adequately. If not, we would be happy to discuss and update our paper further.
> > > > > > >
> > > > > > > Regards,
> > > > > > >
> > > > > > > the authors.

---

### Author Response · Authors · 2023-11-17
**General Reply to All Reviewers**

**Revision summary**

We sincerely thank all the reviewers for their valuable feedback. We have made our best efforts to incorporate all questions, suggestions, and comments into the revised version of our paper.  For the parts we modified in the existing subsection, we have highlighted the changes in blue. As for newly added subsections, only the title is highlighted. We declare the modified part followings:

- **Section 3.1.** Change the expression more rigorously.

---------------------------------------

- **Appendix B.1.** Add concurrent research as related work.

- **(new) Appendix B.4.** Add time-dependent density ratio in GANs as related works.
---------------------------------------

- **Appendix C.** More explanations on overfitting.
---------------------------------------

- **(new) Appendix D.5.** Computational cost
---------------------------------------

- **Appendix E.4.** Effects of scaling on time-independent density ratio.

- **(new) Appendix E.5.** Effects of discriminator accuracy.

- **(new) Appendix E.6.** Comparison to guidance method

- **(new) Appendix E.7.** Objective function interpolation

- **(new) Appendix E.8.** Fine-tuning Stable Diffusion

- **(new) Appendix E.9.** Data augmentation with Stable Diffusion

---

### Author Response · Authors · 2023-11-21
**A kind reminder for further discussion.**

Dear Reviewers,

We express our heartfelt appreciation to the reviewers for their valuable feedback, which we deem essential for enhancing the quality of our work. Kindly request reviewers to examine our responses, as we are enthusiastic about resolving any additional concerns or queries and participating in a meaningful discussion.


Unlike previous years, there will be no second stage of author-reviewer discussions this year. As the deadline for the discussion is approaching, please inform us of any additional questions. Thank you once more for your time and thoughtful consideration.

Best regards,

Authors.

---

### Meta-Review · Area_Chair_CaGG · 2023-12-08

**Metareview:**

This submission was reviewed by four knowledgeable referees. The initial concerns raised by the reviewers included:
1. The unclear practical significance of the proposed approach (EYEW)
2. Baselines and comparisons with fair diffusion missing (EYEW, irEP)
3. Claimed advantages did not appear sufficiently rigorous (EYEW)
4. Limitations carried over from previously introduced techniques (WcK7)
5. Computational complexity was not discussed (irEP)
6. The effect on the generation quality was not discussed (WcK7)

Moreover, the reviewers requested some clarifications and raised additional minor questions.

The rebuttal addressed most of the reviewers concerns by arguing that text-to-image models also suffer from bias, performing the missing comparisons, adding a computational complexity analysis and pointing the reviewers to the tables in the paper which discussed the generation quality. The rebuttal also clarified the doubts raised w.r.t. claims and highlighted how the proposed approach differs from previous works (and improves upon previous limitations). After rebuttal, two reviewers are satisfied with the authors' responses, and two reviewers remain unresponsive. The authors also note the unresponsiveness of these two reviewers and explain how they have attempted to address their concerns. After discussion with the reviewers, the AC agrees that the concerns raised by all reviewers appear to have been addressed. Therefore, the AC recommends to accept and encourages the authors to include the reviewers' feedback in the final version of their manuscript.

**Justification For Why Not Higher Score:**

Although the contribution is relevant, the datasets considered are relatively toy-ish and the impact could have been higher if conditional models were considered.

**Justification For Why Not Lower Score:**

The contribution appears relevant and well executed.

---

### Decision · Program_Chairs · 2024-01-16

Accept (poster)